# Neighborhood-level disparities and subway utilization during the COVID-19 pandemic in New York City

Daniel Carrión [1✉], Elena Colicino [1], Nicolo Foppa Pedretti[1], Kodi B. Arfer[1], Johnathan Rush [1], Nicholas DeFelice[1,2] & Allan C. Just [1,2✉]

The COVID-19 pandemic has yielded disproportionate impacts on communities of color in New York City (NYC). Researchers have noted that social disadvantage may result in limited capacity to socially distance, and consequent disparities. We investigate the association between neighborhood social disadvantage and the ability to socially distance, infections, and mortality in Spring 2020. We combine Census Bureau and NYC open data with SARS-CoV-2 testing data using supervised dimensionality-reduction with Bayesian Weighted Quantile Sums regression. The result is a ZIP code-level index with weighted social factors associated with infection risk. We find a positive association between neighborhood social disadvantage and infections, adjusting for the number of tests administered. Neighborhood disadvantage is also associated with a proxy of the capacity to socially isolate, NYC subway usage data. Finally, our index is associated with COVID-19-related mortality.

[1] Department of Environmental Medicine and Public Health, Icahn School of Medicine at Mount Sinai, New York, NY, USA. [2] Institute for Exposomic Research, Icahn School of Medicine at Mount Sinai, New York, NY, USA. ✉email: daniel.carrion@mssm.edu; allan.just@mssm.edu

The 2019 novel coronavirus (SARS-CoV-2) was first detected in Wuhan, China, and has since become a worldwide pandemic. In the United States, given the nature of this novel infectious disease, anyone exposed to the pathogen was believed susceptible to infection. By the Spring of 2020, there were no proven pharmacologic treatments, and limited testing capacity contributed to a poor understanding of viral transmission. Pre-existing conditions are known risk factors of disease severity, and mortality increases sharply with age[1]. Consequently, the United States federal, state and local governments have principally relied on non-pharmaceutical interventions such as social distancing and mask-wearing. New York State (NYS) on PAUSE is one such effort, whereby essential workers, i.e., healthcare workers, food purveyors, bank tellers, etc., were the only employees that should be reporting to work. We examine the association between social factors, such as employment and commuting patterns, population density, food access, socioeconomic status and access to healthcare, and area-level infection rates.

It has been widely noted in popular media and emerging scientific evidence that COVID-19 is taking a disproportionate toll on communities of color[2–5]. For example, in Chicago, as the outbreak first unfolded, Black people comprised 70% of early COVID-related deaths, but only 30% of the population[5]. In New York City (NYC), Hispanics and LatinX people, and Black people were disproportionately impacted, with mortality rates of 264 and 249 per 100,000 respectively, compared to 124 for white people[5]. While differences in disease severity are likely attributed to higher levels of preexisting conditions, i.e., health disparities[6], this does not explain differences in disease incidence. A survey of laboratory-confirmed hospitalized cases across 14 states in March 2020 found that where race or ethnicity was reported, that 33.1% of hospitalized patients were non-Hispanic Black people[7]. In NYC, as of 13 May 2020, the cumulative incidence of non-hospitalized positive cases were 798.2, 684.8, and 616.0 per 100,000 for Black/African American, Hispanic and LatinX, and white people respectively[8].

A body of literature on the social determinants of health suggests that there are numerous inequities that provide the scaffolding for increased COVID-19 infection rates in communities of color. Racism operates on both the interpersonal and structural levels, the latter explaining the societal mechanisms that reinforce inequality, including through housing, employment, earnings, benefits, health care, criminal justice, etc.[9]. Those structural forms of social disadvantage are responsible for many of the health disparities we observe in communities of color[10].

Researchers have outlined the ways in which residential segregation and structural disadvantages lay the groundwork for racial disparities in infectious diseases[11]. More recently, others have noted that social distancing is more difficult for communities of color[5]. Taken together, this literature highlights the social mechanisms that may facilitate viral spread in communities of color. The underlying structural disadvantages relevant to the current coronavirus pandemic might include that people of color (POC) are more represented amongst low-wage jobs[12], many of which are now deemed essential[13]. When they get home from work, they are more likely to return to densely populated homes and neighborhoods[14]. Further, due to structural or cultural factors, multigenerational homes are more common in communities of color[15], which makes social distancing between least susceptible (healthy children) and most susceptible (older adults with chronic conditions) difficult. POC often live further from supermarkets and sources of nutritious foods, necessitating further travel for groceries[16]. These factors, among others, underscore the many ways that the capacity to social distance may be contextual and based on structural factors.

In this population-level study, we use socioeconomic data on neighborhood characteristics to understand differences in infection incidence between neighborhoods, as we quantify the relative contribution of these measures of social disadvantage and if a proxy of social isolation, NYC subway utilization, helps us to understand these differences. We create a ZIP code level COVID-19 inequity index for NYC, a composite measure of neighborhood-level disadvantage trained on infection rates, and show how this index explains racial/ethnic disparities in cases, thus reflecting structural forms of disadvantage. Finally, we examine the relationship between the inequity index and neighborhood-level COVID-19 mortality. Ultimately, we create a tool that identifies social factors that are associated with viral spread, and therefore, may be useful throughout the US to pinpoint potential areas for targeted public health intervention. The inequity index was designed to understand the relationship between social inequality and neighborhood infection rates during the first wave of the COVID-19 pandemic in New York City. All results are associational and based on population-level, rather than individual-level, data. It is a retrospective tool, and, in its current form, may be used in scientific research for studies with contemporaneous and co-located data. The inequity index cannot, and was not intended to, predict dynamic infection rates or spatial clusters. We caution users of the index to avoid applications that stigmatize neighborhoods or their residents.

## Results

**Cross-sectional COVID-19 inequity index.** We wanted to identify an association between a neighborhood social disadvantage composite index and cumulative COVID-19 viral swab-confirmed infection incidence. There were 174,614 positive tests across 177 NYC modified ZIP Code Tabulation Areas (ZCTAs) as of May 7, 2020. Kendall's tau correlations between social disadvantage variables ranged from −0.15 to 0.61 (Supplementary Fig. 1). Kendall's tau correlation tests were also conducted between each variable and the infection incidence (Supplementary Table 1). Our a priori hypothesis was that increased disadvantage is associated with higher infections, so we transformed variables that had univariate negative associations with the outcome to aid in interpretation. Median income was transformed using its reciprocal, and for proportion-based variables, we used 1—the value.

The BWQS regression analysis identified evidence of an association between our composite variable of ZCTA-level neighborhood social disadvantage (on a ten-unit scale) and the number of infections per 100,000 (Fig. 1) when adjusted for a smooth function of ZCTA-level testing (Supplementary Fig. 2). We found that each unit increase in disadvantage is associated with an 8% increase in infections per capita (risk ratio: 1.08; 95% credible interval: 1.06, 1.09), and the BWQS regression had an overall Bayesian $R^2$ of 0.93 (95% credible interval: 0.92, 0.95)[17] with no significant difference of the observed residuals from the expected distribution (Supplementary Fig. 3). All ten included variables contributed to the composite COVID-19 inequity index, but they did not all contribute equally (Fig. 2 and Supplementary Table 2). We interpreted the point estimates of the weight assigned to the social variables, noting that the credible intervals of the variable weights overlap with one another. We found that the proportion of uninsured people in a ZCTA is the single largest contributor to the impact of social disadvantage on infections, followed by the average number of people in a household and the proportion of the population who are essential workers using personal vehicles to commute. The proportion of uninsured people and the average household size had the highest weights in 39 and 25% of model iterations respectively (Supplementary

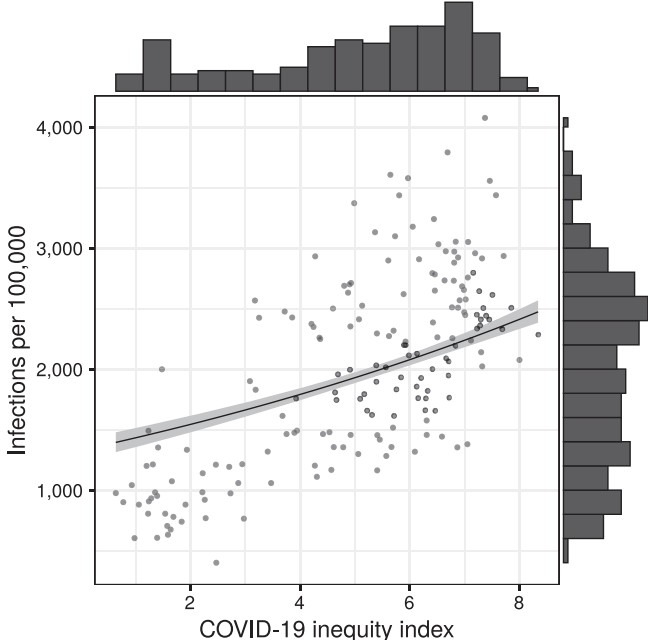

**Fig. 1 Scatterplot of COVID-19 inequity index and cumulative infection incidence.** The unit of analysis is ZCTA ($n = 177$). The fitted line and gray ribbon represent the BWQS negative binomial regression line and its 95% credible interval, holding the testing ratio constant at the median. Marginal histograms represent the distribution of the variable on each axis. Source data are provided as a Source Data file.

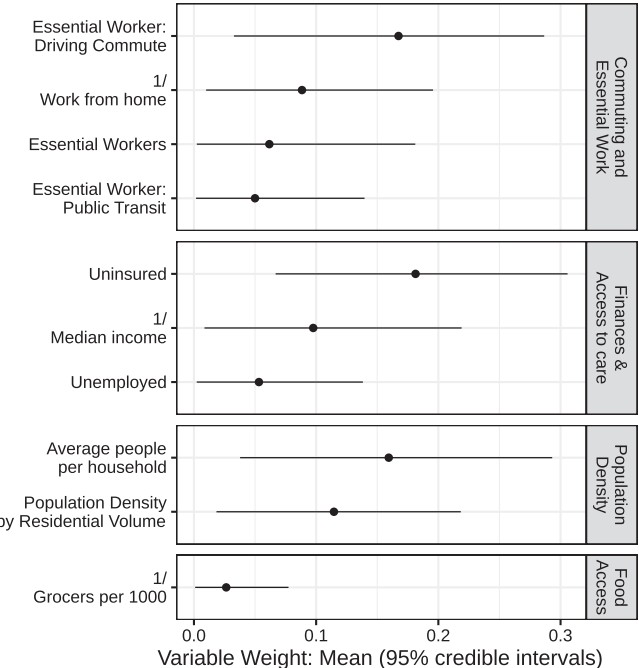

**Fig. 2 Estimated contribution of social variables to the COVID-19 inequity index, with 95% credible intervals.** The unit of analysis is ZCTA ($n = 177$). The BWQS weights the explanatory variables by their relative contribution to the composite index, between 0 and 1. The mean weights sum to 1 and are organized into conceptual domains and ordered by mean weight. Points represent the mean weights and lines represent 95% credible intervals. Source data are provided as a Source Data file.

Table 3). The population density and the median income are also relatively informative compared to the other variables. Since the BWQS has never been applied to social or infectious disease epidemiology, we compared it to two other approaches: (1) a model with only the proportion of essential workers and the median income, and (2) a principal component regression of all ten social variables. The BWQS yielded a smaller root mean squared error and a higher Kendall's tau rank correlation of ZCTAs expected versus observed infections per capita, though the BWQS is a more complex approach (Supplementary Table 4). We also assessed the sensitivity of our results to the selection of alternative priors for our overdispersion parameter and found negligible differences in the coefficients and widely applicable information criterions (WAICs) using half-Cauchy or inverse gamma distributions.

ZCTAs are often not ideal for health-based research, as they may combine heterogeneous neighborhoods with large variation in social phenomena[18]. Infection and mortality data is only publicly available at the ZCTA level, but it is possible that sub-ZCTA geographies more adequately capture the relationship between social disadvantage and infection rates. Given that the mean of each social variable across the entire ZCTA may mask underlying disparities and the social conditions of the more disadvantaged neighborhoods, we repeated our analyses after estimating the median and third quartile of the social variables under study for each ZCTA population using the American community survey (ACS) data from underlying census tracts ($n = 2167$ within NYC). Results showed consistency in the effect estimates and small but notable improvements in the WAIC, Bayesian $R^2$, and RMSE (Supplementary Table 5). We proceeded with the ZCTA-level analysis for ease of interpretation, using ZCTA-level data for the exposure and outcome data.

The spatial distribution of the COVID-19 inequity index largely mirrors that of infections in NYC (Fig. 3). We examined the population demographics of neighborhoods according to their COVID-19 inequity index (Fig. 4). The data shows that Black people have the highest population-weighted mean index and white people have the lowest. Examining these distributions by quantile of the COVID-19 inequity index shows that white populations are overrepresented in ZCTAs in the lower quartile of the COVID-19 inequity index (<25th percentile) and under-represented in the upper quartile of inequity index (>75th percentile) ZCTAs (Supplementary Fig. 4). While white people comprise approximately 32% of NYC's population, they only make up 10% of high inequity index ZCTAs. Conversely, Black people and Hispanic, LatinX people are 22 and 29% of NYC's population and 30 and 43% of high index areas respectively.

**Capacity to social distance.** We used area-level subway ridership as a proxy for the capacity to socially distance, and to examine differences in ridership by our COVID-19 inequity index, thus representing a possible mechanism between neighborhood disadvantage and infections. Given that neighborhood disadvantage is spatially heterogeneous, associated ridership differences may indicate places where individuals were less able to socially distance themselves due to being an essential worker, not having access to a car, etc. We found that capacity to social distance appears lower in higher COVID-19 inequity index areas, as indicated by the most important variables in our BWQS regression analysis. To assess whether or not this was true using longitudinal data, we decided to model differences in subway utilization by united hospital fund areas (UHFs) in NYC. We only included the 36 UHFs with the most consistent data quality and that had subways present (Supplementary Fig. 5). In order to

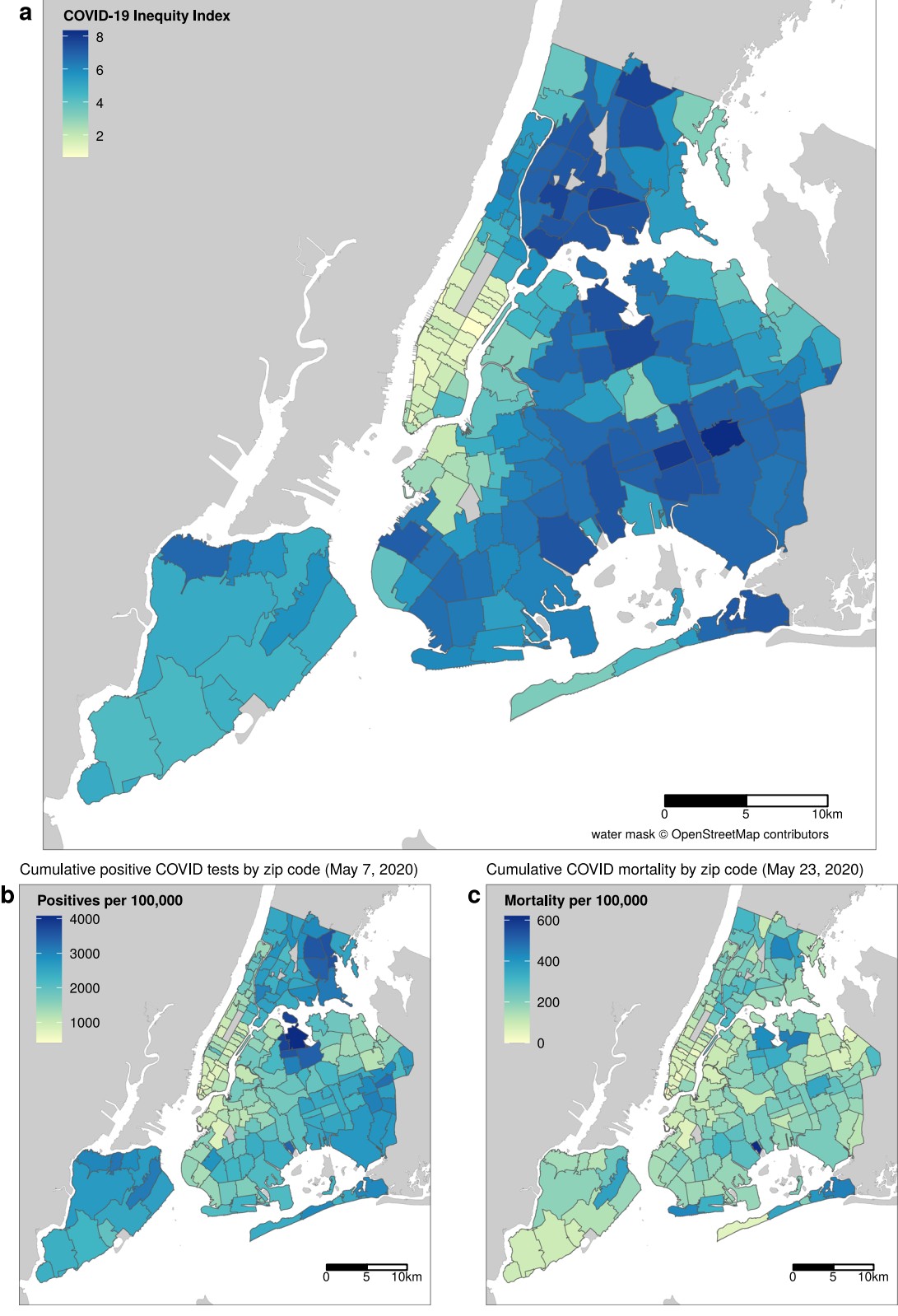

**Fig. 3 COVID-19 inequity index map of NYC in relation to infections and mortality.** The unit of analysis is ZCTA ($n = 177$). **a** Shows the constructed COVID-19 inequity index based on weighted social determinants trained on (**b**) reported cases of SARS-Cov-2 infections as of May 7, 2020. **c** Shows the ZCTA-level mortality per 100,000 as of May 23, 2020. Base map and data from OpenStreetMap and OpenStreetMap Foundation. Source data are provided as a Source Data file.

identify the proper functional form of our nonlinear model, we fit it on the mean sigmoidal decay of subway utilization across all of NYC (Supplementary Fig. 6). We then compared this model to two models, split by UHF-level population-weighted COVID-19 inequity index (Fig. 5). A partial *F*-test demonstrated that the models split by COVID-19 inequity index categories (above versus below the median) were a significantly better fit than one combined model ($p < 0.0001$).

The separate models indicate that there is little difference between slopes for the high (−5.6% per day; 95% CI: −6.0, −5.3%) versus low (−6.2% per day; 95% CI: −6.5, −5.8%) COVID-19 inequity index areas (Table 1). However, the lower asymptote of subway utilization under social distancing policies is higher for high inequity index (16%; 95% CI: 15.3, 16.7%) areas

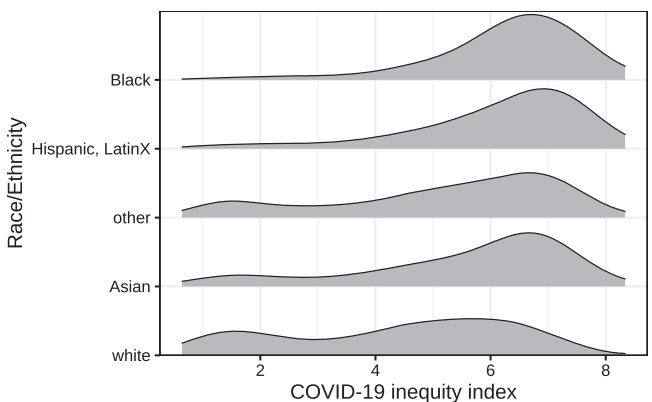

**Fig. 4 Distribution of COVID-19 inequity index by race/ethnicity of ZCTA residents.** The COVID-19 inequity index varied by race/ethnic categories according to the 2018 ACS. Density plots are population-weighted to demonstrate the relative abundance of representation according to ZCTAs and their corresponding COVID-19 inequity index and ordered by the population-weighted mean index. Source data are provided as a Source Data file.

compared to low-risk inequity index areas (9.5%; 95% CI: 8.9, 10.1%). This implies that high risk and low index areas had similar relative rates of decreased subway utilization upon news of the pandemic, e.g., school closures, etc. However, high COVID-19 inequity index neighborhoods had higher relative use of the subway system after official social distancing policies (NYS on PAUSE) went into effect. These trends were consistent when we modeled 3 risk groups at the UHF (Supplementary Fig. 7) and ZCTA levels (Supplementary Fig. 8). Overall subway utilization followed similar trends to other measures of transportation, including Google's transit data (see Supplementary Fig. 9).

**Mortality related to the COVID-19 inequity index.** There were 16,289 COVID-related deaths across 177 ZCTAs by May 23, 2020. NYC DOHMH surveillance data show that race/ethnic disparities are greater for COVID-19 mortality than for SARS-Cov-2 infections. Therefore, we wanted to assess whether our index that captures neighborhood disadvantage in relation to infection was also related to ZCTA-level mortality. Results from the negative binomial model show a strong association between the ZCTA COVID-19 inequity index and cumulative COVID mortality incidence (Table 2). This regression model employed a spatial filtering approach to account for potential spatial auto-correlation at the ZCTA level. We found that each unit increase in the COVID-19 inequity index is associated with a 20% increased risk of COVID-related mortality (relative risk: 1.2; 95% CI: 1.16, 1.23) when accounting for spatial dependence. Spatial autocorrelation of residuals was small in magnitude (Moran's I: 0.05) and non-significant (*p* value: 0.08). See Supplementary Fig. 10 for a map of the residuals.

## Discussion

We conducted an ecological study using publicly available data to identify associations between neighborhood social disadvantage on cumulative COVID-19 infections and COVID-19-related mortality in NYC over 9 weeks after the first COVID-19 case was identified in spring 2020. The COVID-19 inequity index was

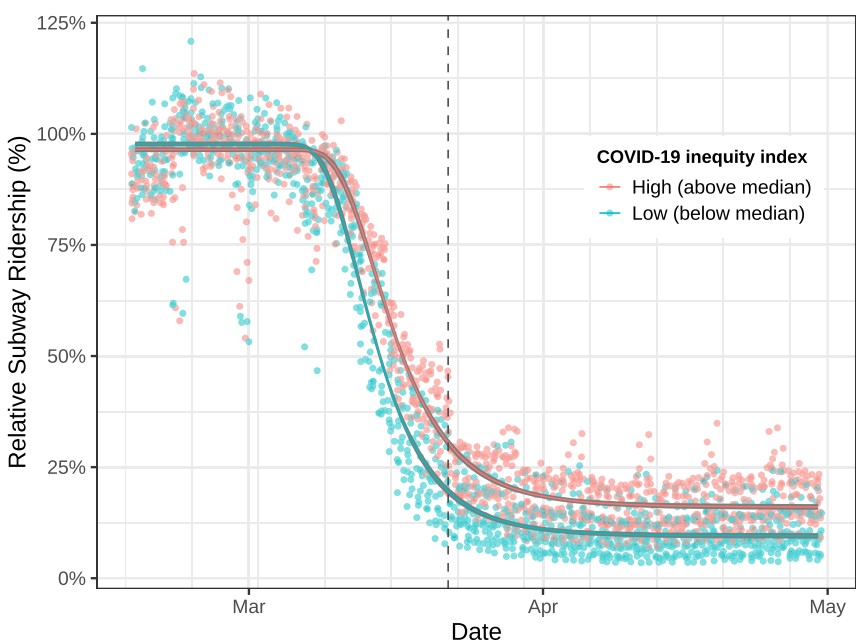

**Fig. 5 Subway ridership trends by population-weighted COVID-19 inequity index at the united hospital fund neighborhood level.** The nonlinear model was fitted using a generalized Weibull equation with two curves: high (above median) and low (below median) COVID-19 inequity index at the UHF neighborhood level (*n* = 36). Daily subway ridership is relative to 2015–2019. The dashed line represents the start of NYS on PAUSE social distancing policies. Ridership is shown between 16 February 2020 and 30 April 2020. Source data are provided as a Source Data file.

**Table 1 Coefficients from nonlinear regression of COVID-19 inequity index and subway ridership.**

|  | High inequity index | | Low inequity index | | |
|---|---|---|---|---|---|
|  | Coefficient | 95% CI | Coefficient | 95% CI | P value |
| Lower asymptote | 16% | 15.3, 16.7% | 9.5% | 8.9, 10.1% | $2.00 \times 10^{-16}$ |
| Slope | −5.7% per day | −6.0, −5.3% | −6.2% per day | −6.5, −5.8% | $4.50 \times 10^{-1}$ |

Lower asymptote and slope are parameters from a nonlinear model with a generalized Weibull equation and two curves for high (above median) and low (below median) COVID-19 inequity index at the UHF neighborhood level. Two-tailed tests, no adjustments for multiple comparisons.

**Table 2 Results from a negative binomial regression of COVID-19 inequity index and COVID-19 cumulative mortality proportion at the ZCTA level.**

| Term | Relative risk (95% CI) | P value |
|---|---|---|
| COVID-19 inequity index | 1.2 (1.16, 1.23) | $2.00 \times 10^{-16}$ |
| Eigenvector of spatial element | 0.41 (0.19, 0.86) | $2.00 \times 10^{-16}$ |

also used to understand differences in social distancing, as measured by subway ridership. In creating our COVID-19 inequity index, we found that a combination of social variables, indicative of social disadvantage, is associated with cumulative infections and mortality. Black communities and Hispanic, LatinX communities are overrepresented in high COVID-19 inequity index neighborhoods, and white people are overrepresented in low COVID-19 inequity index neighborhoods, which may represent structural forms of racism. When examining differences in the capacity to socially isolate, we found that high index neighborhoods had higher subway ridership during NYS-mandated social distancing. Finally, our COVID-19 inequity index is also associated with cumulative COVID-19 mortality at the ZCTA level. This implies that the same social factors that inform increased disease risk are also associated with severe outcomes, either directly or through intermediates.

A growing body of literature is examining the greater impact of COVID-19 based on measures of neighborhood structural disadvantage. As some have noted, COVID-19 is not creating new health disparities, but exacerbating those that already exist[2]. A recent investigation found that county and ZCTA area-based socioeconomic measures, specifically using crowding, percent POC, and a measure of racialized economic segregation, were useful in identifying higher COVID-19 infections and mortality in Illinois and New York[19]. Work on COVID-19 mortality in Massachusetts has found excess death rates for areas of higher poverty, crowding, proportion POC, and racialized economic segregation[19]. There has been some concern that neighborhood disadvantage and case positivity associations could be confounded by access to testing, with advantaged communities having more access than lower[20]; however, Schmitt-Grohe et al. found no evidence of testing inequalities by income, and a small negative association when also including the proportion of Black people[21]. An analysis examining spatial patterns of infections found that ZCTAs with high proportions of Black residents and residents with chronic obstructive pulmonary diseases (COPD) were the most likely to experience the highest COVID-19 infection rates[22]. We know of three other studies that have tied neighborhood-level disadvantage to mobility (via cellphone data)[20] and subway utilization[23], and mobility data with COVID-19 infections[20,23,24]. Researchers have begun to identify counties that are particularly susceptible to severe COVID-19 outcomes using a combination of biological, demographic, and socioeconomic variables[19]. They identified counties with high population density, low rates of

health insurance, and high poverty as particularly at risk. However, a stated limitation of this work is that many of these variables are interrelated.

Our study has many strengths. First, we acknowledge and address the strong interrelation of social variables by using a data-driven method for modeling mixtures of exposures: BWQS, while flexibly adjusting for the intensity of testing. By using this method, we create a composite index that captures the combined effect of the constituent variables after a quantile transformation that makes our model more robust to extreme values. This process is also supervised, meaning that the variables are not weighted equally in the composite index, but instead, the approach empirically learns their individual contributions to explaining the outcome. The appeal of an index is both summative and contextual. The composite index can be used to direct outreach and testing (such as mobile testing units) regardless of the components most driving the index value in each geographic unit. Also, index weights can be used to inform the types of interventions targeted to specific neighborhoods. In NYC, COVID-19 positive residents could stay in hotels for two weeks until they were no longer contagious, but it is unclear if this was a targeted intervention. Our index helps contextualize the unique combination of social conditions per ZCTA that relate to ZCTA-level infections for targeting interventions (e.g., in ZCTAs with the higher average number of people per household, then rapid testing could be paired with outreach for hotel isolation). The second strength of our study is that our approach largely relies on ACS data, which is available across the USA, and may allow for the identification of other communities nationwide that are particularly vulnerable to future outbreaks or even other novel respiratory pathogens. Third, we explicitly excluded race and ethnicity from the creation of the index because we were more interested in identifying social processes associated with infection rates, rather than those that may imply biological or behavioral explanations to health disparities[25]. The theory underlying these relationships is that structural racism increases the proportion of POC in areas of high disadvantage, and those structural forms of disadvantage facilitate pathogen spread. To demonstrate this, we employed the index to understand neighborhood differences in the capacity to social distance. This finding provides additional evidence that low-income communities and communities of color may be less able to socially distance[5]. Fourth, our spatial analysis of COVID-19-mortality shows that the COVID-19 inequity index may not only be useful in identifying infection risk, but also a risk of severe outcomes. Finally, our data sources and analysis code are publicly available, which we believe is one of few among comparable COVID-19 disparities analyses[24]. This means that others can (1) reproduce these analyses, (2) expand on the work by assessing different modeling strategies, and (3) assess the utility in other parts of the country.

This study also has notable limitations. First, we were unable to identify a measure of multigenerational housing at the ZCTA level, which may represent a pathway for infection, and potentially severe disease. Second, by not including race in our models,

we may be missing an opportunity to tune these models to the impacts of interpersonal and structural forms of racism[26]. Third, infection data are based on viral swab-confirmed cases, and early testing protocols in NYC were largely limited to hospitalized individuals, therefore those with more severe disease[8]. Consequently, ZCTA infection data may be confounded by the distribution of factors that drive disease severity. We addressed this by adjusting our BWQS regression for the amount of overall testing per ZCTA. Relatedly, for our spatial analysis of COVID-mortality, we were unable to access a ZCTA-level measure of chronic diseases. Since communities of color have higher rates of chronic disease at younger ages[27], and chronic diseases increase the likelihood of severe COVID-19 outcomes, this is an important challenge. However, because social disparities are a major contributor to differences in the chronic conditions that increase the likelihood of severe disease, we did not want to adjust for an effect modifier. Instead, we adjust for spatial autocorrelation to account for residual risk factors that are more similar in nearby neighborhoods. Fourth, we use pre-pandemic social variables derived from the 2018 ACS and thus do not directly account for variation in residential mobility during NYS on PAUSE, i.e., those who fled to their second homes[28]. However, this should be captured, in part, by median income and other measures of affluence in our COVID-19 inequity index. Fifth, our analysis of public transit only utilized data from subway turnstiles, but not bus ridership. Although buses are an important form of transit in NYC, especially in the outer parts of the boroughs, the MTA does not provide time-varying ridership data. Further, buses were made free during the pandemic, so accurate ridership data are likely unavailable to the NYC government as well[29]. Sixth, the BWQS method has never been applied to social/infectious disease epidemiology, although it has been successfully used in environmental epidemiology, which has similar issues of correlated multi-dimensional exposures[30,31]. Seventh, this study uses ZCTAs as our administrative unit of analysis. There is likely greater demographic heterogeneity in some ZCTAs compared to others, and few health-related decisions are made on this level. However, the NYC and NYS DOH has only released data at this scale, thus limiting our ability to examine relationships at varying spatial scales. We assessed the sensitivity of our results to using tract-level data to estimate our ZCTA-level exposure metrics and found consistent results. Finally, an unfortunate potential consequence of creating a COVID-19 inequity index is the possibility of stigmatization of neighborhoods with high index values[25]. This is not our intention, and hopefully not the effect, as our goal is to identify social factors associated with viral spread and demonstrate that uniform mandates on social distancing to avoid exposure are not equally observable by all populations within NYC.

Our work focused on the social conditions that are associated with uneven exposure to SARS-CoV-2, and higher infections, for disadvantaged communities in the spring of 2020. Since then, this situation has been compounded by national interventions that do not appear to be yielding equitable outcomes. For example, the federal Paycheck Protection Program is largely not accessible to minority-owned businesses[32] and bias has been found in the disbursement of Coronavirus aid, relief, and economic security act funding, with hospitals in predominantly Black communities receiving fewer dollars on average[33]. These examples underscore an important reality; addressing health inequities requires explicit identification of disparities, their derivation, and creating interventions designed for equitable outcomes. This is essentially the strategy behind the National Academies of Sciences, Engineering, and Medicine's Consensus Report on the allocation of COVID-19 vaccines, which proposes targeted distribution based on measures of social vulnerability[34]. Consequently, we believe that tailored health and social equity initiatives represent an important path forward for COVID-19 pandemic response and planning.

In this study, we created a neighborhood measure of social disadvantage that is specifically tuned to the impacts of COVID-19 infections and mortality in NYC and we show that this measure is associated with the capacity to socially distance, which may represent an important pathway for COVID-related health disparities. This is an important area of investigation given the large toll that COVID-19 has had, and will likely continue to have unless action is taken, on disadvantaged communities of color in NYC and elsewhere. Future work should assess the generalizability of these results in other parts of the country, new waves of the pandemic, or if our approach can be adapted to different contexts, potentially yielding regionally tuned sets of social variables that are associated with increased COVID-19 inequities.

## Methods

**Data sources and cleaning.** SARS-CoV-2 testing and COVID-19 mortality data. The NYC Department of Health and Mental Hygiene (NYC DOHMH) has been publicly releasing daily testing data (positive and total tests) at the patient's home ZIP Code Tabulation Area (ZCTA) level since April 1, 2020, and COVID-19 related mortality data since mid-May, both available on GitHub[35]. The NYC DOHMH utilizes modified ZCTA geographies, designed to still be mergeable to the Census Bureau ZCTA designations. Our analyses relied on pre-pandemic demographic data to describe variation in neighborhood-level disease burden after much of the community had the potential for exposure. Since spatiotemporal infection patterns were highly variable at the beginning of the pandemic in relation to many independent viral introductions within NYC[36], we estimated cumulative infections on 7 May 2020, 4 weeks after NYC's peak infection period. We estimated the time from symptom onset to death as 16 days[35]. Therefore we chose 23 May 2020, for our cumulative COVID-19 mortality analysis. This analysis is not human subjects research as it did not include any intervention or interaction with individuals or any identifiable private information.

Census data. We downloaded the Census Bureau's 2018 ACS data via the tidycensus R package[36]. Data were collected for the 214 ZCTAs in NYC and summarized to the 177 modified ZCTAs that NYC DOHMH reported. Variables included: the total population, number of households, median income, median health insurance status, unemployment, individuals at or below 150% of the federal poverty level, race and ethnicity, industry of employment, and mode of transportation to work. A full list of variables is provided in Supplementary Table 6. We created a proxy for the proportion of individuals in essential worker positions using industry of employment variables. This estimate of essential workers was a sum of those who reported employment in the agricultural, construction, wholesale trade, transportation and utilities, and education/healthcare industries, divided by the total working-age population. To account for teachers mostly working from home, and healthcare workers being essential, we included only half of the education/healthcare industry respondents. From these data, we also estimated the average housing burden by dividing the average rent by median income and the household size by dividing the total population by the number of households. We utilize race and ethnicity according to the following categories: Non-Hispanic Asian, Non-Hispanic Black, Non-Hispanic white, Hispanic/Latino of any race, and aggregate all other races into Other people.

Residential buildings and food access data. We calculated the volume of residential space by merging the NYC building footprints dataset and the primary land use tax lot output dataset. We divided residential volume by total population to calculate mean residents per residential volume, a metric of residential population density. Food access was used as a measure for the likelihood that individuals need to leave their neighborhoods for basic necessities. We estimated food access using data from New York State's Open Data portal for retail food stores. Businesses were restricted to J, A, and C establishment code designations in order to identify those most likely to provide fresh foods and produce, and then manually removed any business names that indicated being a corner store or pharmacy, or primarily selling alcohol/tobacco. The remaining stores had 1034 unique addresses, and we were able to geocode 997 of these to point locations. We spatially joined the point locations to our ZCTA shapefile and divided them by the total Census population to calculate a "grocers per 1000 people" variable as a proxy for food access.

Mobility and transit data. The metropolitan transit authority (MTA) of NYC releases subway utilization data on a weekly basis. These data include the number of entrances and exits per station. For each day and geographic area, we summed all system entrances and exits. To account for typical usage of the subway each month and day of the week, we divided the total turnstile count for each day and area by the median daily count on the same day of the week within the same month throughout the period 2015–2019.

**Quantitative analyses**. Cross-sectional COVID-19 inequity Index. Socioeconomic variables are known to be closely correlated with one another, which is a challenge to model fitting and interpretation of the underlying latent relationship. Researchers have often addressed the multicollinearity of social determinants with the use of dimensionality reduction techniques such as principal components analysis (PCA) in the case of the neighborhood deprivation index[37]. However, traditional PCA only considers correlations between SES variables, whereas a supervised method captures features most relevant for the outcome. To address these shortcomings, we developed a weighted combination of socioeconomic variables to explain the cumulative number of COVID-19 cases per ZCTA using Bayesian weighted quantile sums regression (BWQS)[30] (see Supplementary Methods for model notation and description). Candidate demographic variables for the COVID-19 inequity index from the ACS included average household size, income, rent, households using supplemental nutrition assistance program (SNAP) benefits, poverty, health insurance, unemployment, and industries of employment. Derived variables from ACS data included average rent burden and household size. Non-ACS variables included the population density (persons per square foot of the ZCTA) and residential population density (persons per cubic foot of ZCTA residential volume). Our choice of variables is largely influenced by the theoretical framework from Acevedo-Garcia[11] and our understanding of the employment sectors deemed essential workers, and therefore less able to stay home during a time of social distancing. Our measures of population density attempt to capture (in)ability to physically distance within the home and otherwise dense housing conditions such as apartment buildings versus single-family homes, and therefore higher risk of contact with infected individuals. Finally, SNAP benefits and measures of food access are included to indicate further travel from home for basic necessities and/or less opportunity to amass food reserves to reduce the overall frequency of shopping. We restricted candidate variables to those that would be realistically publicly available or accessible in other parts of the United States. A directed acyclic graph of the proposed relationship between variables is depicted in Supplementary Fig. 11.

The BWQS distinguishes two groups of predictors. In one group, which comprises our socioeconomic variables, the predictors are transformed into quantiles using the empirical cumulative distribution function to limit the influence of outliers and multiplied by ten to allow the COVID-19 inequity index to range from [0, 10]. The variable weights are forced to lie in [0, 1] and sum to 1 with a uniform Dirichlet prior. We included a large candidate list of socioeconomic variables in the BWQS, and all posterior probability distributions were estimated leveraging a Hamiltonian–Monte Carlo algorithm, which is an efficient algorithm for reducing correlation between sample states and improving the stability of the variable weight estimates. Model diagnostics were monitored to evaluate chain convergence and appropriateness of fit for each parameter. Quantile residuals were constructed using posterior draws for diagnostics and visualization[38]. Our final list of variables was based on an iterative process according to: (1) maximizing model fit and penalizing complexity, measured by the WAIC, (2) removing one variable when bivariate correlations were high ($|\tau| \geq 0.9$), and (3) our understanding of underlying social processes in relation to infectious disease. The other group of variables in a BWQS regression is the covariates, which in our case consist solely of a natural spline smoother (3 degrees of freedom) for the testing ratio (the total number of reported tests divided by the population per ZCTA). We included this to account for variation in disease surveillance. The predictor is untransformed and the coefficients are less constrained, using a normal prior with mean 0 and SD 100. A negative-binomial distribution is used for the dependent variable: the cumulative number of positive SARS-CoV-2 tests per 100,000 people. The resulting weighted index was our COVID-19 inequity index. We compared the results of the BWQS model to two other approaches. First, we conducted a negative binomial regression with median income and percent uninsured, adjusting for the testing ratio using a natural spline. Second, we conducted a PCA of the same ten social variables in the BWQS. We took the first component (explaining 41.6% of the variance) and included it in a negative binomial regression, adjusting for the testing ratio using a natural spline. The model predictions were compared to the BWQS results using the mean absolute error and Kendall's tau metrics. Kendall's tau was specifically used to assess the rankings of infections per capita between ZCTAs, as that would likely be more important to public health practitioners needing to prioritize interventions.

We visualized the distribution of the COVID-19 inequity index values by self-reported race/ethnicity as per the ACS categories and total population. We also separate the inequity index values into three categories: below the 25th percentile, between the 25th and 75th percentiles, and above the 75th percentile. Populations were aggregated by race/ethnicity and then divided by the total population of the associated ZCTAs.

Robustness of ZCTA-level measures. ZCTAs vary in size (populations from 3028 to 112,425 from the 2018 ACS) and may combine heterogeneous neighborhoods with large variations in social variables. To evaluate whether the ZCTA-level summary (mean) of social variables we used in constructing our COVID-19 inequity index adequately captures the population distribution, or whether infection rates for a ZCTA population is more closely related to the social variables of those who are more disadvantaged within that ZCTA, we estimated the median and third quartile of the social variables understudy for each ZCTA population using ACS data from census tracts ($n = 2167$ within NYC). Given the many-to-many relationship of non-aligned ZCTAs and census tracts, we

estimated weighted quantiles of each of our social variables using the June 2020 HUD crosswalk tables[39] that attribute the proportion of residential households in each ZCTA living within each overlapping census tract reweighted by the average household size.

Model description for BWQS regression. Here we provide a description of BWQS regression when the outcome variable $Y$ has a negative binomial density distribution. The formulation of the negative binomial density distribution is:

$$NB(y|\mu,\phi) = \frac{\Gamma(y+\phi)}{\Gamma(\phi)\Gamma(y+1)}\left(\frac{\mu}{\mu+\phi}\right)^y\left(\frac{\phi}{\mu+\phi}\right)^\phi \quad (1)$$

where $\mu \in R^+$, $\phi \in R^+$ and $y \in N$.

The mean and variance of a random variable $Y \sim NB(y|\mu,\phi)$ are:

$$E[Y] = \mu \text{ and } \text{Var}[Y] = \mu + \frac{\mu^2}{\phi} \quad (2)$$

We included the negative binomial distribution with $\eta = \log(\mu)$ where $\eta \in R$ in the BWQS regression framework, so that the BWQS regression model has the following form:

$$\eta = \beta_0 + \beta_1 * BWQS + \delta^T X \quad (3)$$

where $\beta_0$ is the intercept; $\beta_1$ is the coefficient mapped to the COVID-19 inequity index of $N_C$ mixture components previously transformed into quantiles and multiplied by ten ($q$); BWQS index is $\sum_{j=1}^{N_C} w_j q_{ij}$ with weight $w_j$ for the $j$-th mixture component; and $\underline{\delta}$ is a vector of coefficients mapped to the $N_k$ covariates $X$ which in our model is a natural spline basis transformation of the testing ratio.

The choice of the prior of the model is based on prior literature, the prior definition of the BWQS regression, and their properties of being conjugate:

$$\begin{aligned} \beta_0, \beta_1 &\sim N(0, 100), \\ \delta &\sim N_{N_k}(0, 100*I_{N_k}), \\ \phi &\sim inv\Gamma(0.01, 0.01), \\ w &\sim Dirichlet(1_{N_c}) \end{aligned} \quad (4)$$

The Dirichlet distribution $Dir(\boldsymbol{\alpha})$ is a multivariate generalization of the Beta density distribution and it belongs to a family of continuous multivariate probability distributions parameterized by a vector $\boldsymbol{\alpha}$.

The $\boldsymbol{\alpha}$ vector has the characteristics of the multinomial parameter, i.e., the components of the $\alpha$ vector ($\alpha_i$ for all $i$-th component) are positive reals and the sum of all components is equal to 1 ($\sum_{i=1}^I \alpha_i = 1$). This second characteristic implies that the estimates of all parameters are not independent, similarly to what we have with the multinomial distribution. For these characteristics, the Dirichlet distribution is commonly used as the prior for the multinomial distribution.

The Dirichlet distribution is also widely used as prior distribution because of its property of being conjugate, which means that the posterior distribution will be a Dirichlet with parameters α different from initial values. For this reason, this distribution has an easy computation and facilitates quantification of how much the prior beliefs have changed after including data in the model. In our case the Dirichlet is parametrized by a parameter vector $\boldsymbol{\alpha} = (1, 1, \dots, 1)$, which assumes a uniform density distribution across the domain, implying a non-informative prior for all weights. Changes in the $\boldsymbol{\alpha}$ parameter vector suggest stronger assumptions about the importance of each variable, which we do not have a priori. In other words, the $\alpha$ parameter vector rules the shapes of the distribution; $\alpha_i = 1$ assumes uniform distribution across the domain of the $i$th mixture component, implying no prior information about its contribution to the overall mixture. The full BWQS package is available on GitHub: https://github.com/ElenaColicino/bwqs.

Alternative priors for the overdispersion parameter $\phi$ include a half-Cauchy distribution or an Inverse-Gaussian distribution. While we chose minimally informative priors, we also conducted a sensitivity analysis in which we used a half-Cauchy (0, 3) distribution for the overdispersion parameter instead of the Inverse-Gamma (0.01, 0.01) in our main model, ultimately comparing the resulting coefficients and WAICs.

Capacity to social distance. Our BWQS model uses cross-sectional data to create a COVID-19 inequity index, but we wanted to assess the degree to which those differences in infections were explained longitudinally by the inability to socially isolate/distance. We could not assess this directly, so we chose to look at subway ridership in relation to the COVID-19 inequity index. We utilized MTA transit data as a proxy for social distancing since public transit utilization during this time period may reflect conditions that contribute to greater exposure risks, including essential work and less ability to socially distance. Subway stations are in a fraction of NYC ZCTAs, and individuals often traverse ZCTAs to get to a station, so we aggregated subway utilization to 42 UHF neighborhoods. UHF neighborhoods are composed of adjacent ZCTAs approximating community districts. Aberrantly low utilization observations (<10%) in February and early March 2020 were removed when explained by planned weekend service changes—specifically those in low subway density areas. We computed a population-weighted COVID-19 inequity index per UHF.

We modeled change in relative subway usage leading up to, and during, the NYS on PAUSE period. Relative subway utilization is a proportion, therefore the transition from business-as-usual to social distancing roughly followed a sigmoidal

decay. A mean nonlinear response can be modeled by nonlinear least-squares when a functional form is specified, as implemented by the *drc* R package[40]. We utilized a generalized Weibull formula, which took the following functional form:

$$relative\ use = c + (d - c)(1 - \exp(-\exp(b(\log(timeindex) - \log(e))))), \quad (5)$$

where $c$ is the lower asymptote, $d$ is the upper asymptote, $b$ is the slope, *time index* is the transformation of the date as an integer, $e$ is the inflection point of the function, and *relative use* is the proportion of subway ridership. We fit two models, one for neighborhoods with a COVID-19 inequity index at or below the median and one for those above the median. We estimated parameters with the maximum-likelihood method. We compared the slopes ($b$) and the lower asymptotes ($c$) of the two models to investigate differences in the ability to socially isolate. To assess the consistency of our results based on administrative units and BWQS cut points, we repeated the analysis using three COVID-19 inequity index groups at the UHF level, and then again at the ZCTA level.

**COVID-19 inequity index and mortality.** Given high COVID-related mortality in disadvantaged communities, we wanted to assess if our COVID-19 inequity index was also associated with cumulative COVID mortality by the total population. To do so, we employed a negative binomial model, regressing ZCTA-level COVID mortality on the COVID-19 inequity index. In order to adjust for spatial autocorrelation, and thus unmeasured spatial confounding, we employed a spatial filtering approach whereby we identify the eigenvector associated with spatial autocorrelation (as measured by Moran's I), and explicitly adjusted for those values in the negative binomial regression[41,42]. The goal, then, was to "filter out" spatial autocorrelation from the residuals. Negative binomial models were implemented with the *MASS* package, supplemented with spatial functions from the *spdep* and *spatialreg* packages[43,44].

**Mapping and coding.** Geoprocessing and visualization of spatial data were conducted with the *sf* package in R[45]. All analyses were conducted in R version 4.0.2[46].

**Reporting summary.** Further information on research design is available in the Nature Research Reporting Summary linked to this article.

## Data availability

Census data were drawn from https://api.census.gov/data/ using the tidycensus package in R. NYC buildings data were drawn from https://www1.nyc.gov/assets/planning/download/zip/data-maps/open-data/nyc_pluto_20v3_csv.zip and https://data.cityofnewyork.us/api/geospatial/nqwf-w8eh?method=export&format=Shapefile. Zip code neighborhood definitions were accessed from https://web.archive.org/web/20210221151212/https://www.health.ny.gov/statistics/cancer/registry/appendix/neighborhoods.htm. NYC COVID-19 testing and mortality data: https://raw.githubusercontent.com/nychealth/coronavirus-data/6d7c4a94d6472a9ffc061166d099a4e5d89cd3e3/tests-by-zcta.csv. United Hospital Fund shapefile: https://www1.nyc.gov/assets/doh/downloads/zip/uhf42_dohmh_2009.zip. NYC Boroughs shapefile: https://data.cityofnewyork.us/api/geospatial/tqmj-j8zm?method=export&format=Shapefile. Modified ZCTA shapefile: https://data.cityofnewyork.us/api/geospatial/pri4-ifjk?method=export&format=Shapefile. Food retailers in New York State: https://data.ny.gov/api/views/9a8c-vfzj/rows.csv. Crosswalk table of ZCTAs to modified ZCTAs: https://raw.githubusercontent.com/nychealth/coronavirus-data/master/Geography-resources/ZCTA-to-MODZCTA.csv. Crosswalk table of ZCTA to Census Tracts: https://www.huduser.gov/portal/datasets/usps/ZIP_TRACT_062020.xlsx. Geocoding tool for New York State: https://gisservices.its.ny.gov/arcgis/rest/services/Locators/Street_and_Address_Composite/GeocodeServer/findAddressCandidates?f=json&maxLocations=1&SingleLine=. Source data are provided with this paper.

## Code availability

All analytic code, including download procedures, are available to the public[47]. A compiled literate programming html report of all code with all generated output is available at https://justlab.github.io/COVID_19_admin_disparities.

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

## Acknowledgements

This work was supported by NIH grants UL1TR001433 and P30ES023515. DC was funded by NIH T32HD049311. Thanks to Sebastian Rowland for his thoughtful comments on a draft. We gratefully acknowledge the OpenStreetMap contributors, who provided the data for the water mask used in Fig. 3 and supplemental Figs. 5 and 10.

OpenStreetMap data is available under the Open Database License. For more information, see www.openstreetmap.org/copyright and opendatacommons.org.

## Author contributions

D.C. and A.C.J. conceptualized the study and D.C. drafted the manuscript. D.C. conducted all analyses with statistical support from E.C. and N.F.P. A.C.J., E.C., and N.D. provided feedback on design and analysis. The Bayesian Weighted Quantile Sum regression was designed and implemented by E.C. and N.F.P., with the log link function implemented by NFP. KBA developed the procedures and indices for relative subway utilization. J.R. ingested DOHMH data, geocoded food retailers, conducted tract level sensitivity analyses, compiled the literate programming html report, and improved analysis reproducibility. All authors reviewed and approved the manuscript.

## Competing interests

The authors declare no competing interests.
