## [Peer Review File · Nature Communications]

REVIEWER COMMENTS

Reviewer #1 (Remarks to the Author):

This paper examines the associations between a social disadvantage index and COVID-19 infections/mortalities and also explores how subway ridership in different neighborhoods changed around the implementation of the 'NYS on PAUSE' social distancing policy. My comments on this paper are below.

1. I am not exactly sure how the social distance/subway ridership analysis fits into the broader narrative of this study. Is it meant to illustrate one possible mechanism linking disadvantage and COVID? If so, please justify why subway ridership is a good variable for representing the capacity to social distance. It is not clear - either from the paper or from my intuition - why zip codes with higher subway ridership would have lower levels of social distancing. Riding the subway is only one of many possible exposure settings for COVID and only one of many possible ways to (not) practice social distancing. Has this variable been validated in other research vis a vis more direct measures of social distancing (e.g., movement data of cell phones or similar)?

2. What is the rationale for creating a social disadvantage index and why is this index better - for representing different types of neighborhood environments and for informing policy - than simply analyzing a smaller set of variables (not transformed into an index)? I understand that socioeconomic variables are often highly correlated and so multicollinearity is a concern, but is multicollinearity actually an issue in this study? Are there theoretical or policy-oriented reasons as to why a single index does a better job capturing neighborhood characteristics than multiple variables?

3. One additional limitation of this study that deserves some discussion is the use of zip code-level data. I can think of two issues in particular. One, zip codes are quite large and are likely internally heterogeneous with respect to the COVID variables and disadvantage. Two, because of this high internal heterogeneity and because zip codes are not the same units of analysis used by health policy professionals, they may not be suitable for designing/implementing policies.

4. Did the authors complete any sensitivity analyses for the prior distributions in the BWQS model? I am particularly interested in if/how the model results may change when the Inverse Gamma(0.01, 0.01) prior for the overdispersion parameter is modified.

5. Are the results presented in Figure 5 sensitive to the choice of two groups (above median BWQS index vs. below median BWQS)? Supp Fig 2 breaks down the racial/ethnic composition in 3 zip code groups, so why does Fig 5 focus on only 2 groups? Is it possible that Fig 5 may show a different pattern if the

ridership curves are fit to 3 or 4 groups?

6. Under 'Capacity to Social Distance' in the Methods section, the authors note that they want to 'asses the degree to which infections were explained by inability to socially isolate/distance'. But the analysis that addresses this statement focuses on BWQS index values and not COVID infection rates, so there appears to be some inconsistency in the language here.

7. It would be helpful if the authors could provide formal notation of the BWQS model in the Supplementary Material.

8. Please include a map of the model residuals in the Supplementary Materials to support the claim that there was no residual spatial autocorrelation. A visual inspection of the model residuals is just as informative as the Moran's I diagnostic.

9. Add a scale bar to the maps. This will help readers who are unfamiliar with NYC zip codes to interpret the size of the areas analyzed.

10. Please define the acronym UHF when it is first introduced in the paper. I think that this may be an issue with my version of the paper as the data section is after the discussion section.

11. While it is hard to keep up with the new papers in this field, the results presented here are original to the best of my knowledge. Regardless, I think that it would be prudent to add some discussion as to how this study is similar/different from other studies looking at zip code COVID data. Specifically, (a) Chen and Krieger 'Revealing the unequal burden...', (b) DiMaggio et al. 'Blacks/African Americans are 5 Times More Likely to Develop COVID-19', (c) Borjas 'Demographic Determinants of Testing Incidence', and (d) Schmitt-Grohé et al. 'COVID-19 Testing Inequality in New York City'

Reviewer #2 (Remarks to the Author):

In this paper, the authors address an important topic and seek to assess the role of neighborhood-level structural disadvantage on observed racial/ethnic inequalities in COVID-19 cases and deaths as well as the ability for disadvantaged groups to socially distance. The paper presents the results of three main analyses. (1) A risk index is created using Bayesian weighted quantile sums regression on 10 neighborhood-level covariates and positive tests. The authors find that a one unit increased in this risk index is associated with a 110% increase in infections per capita. (2) Using subway ridership data, the

authors compared the high risk neighborhoods (> median risk) to low risk neighborhoods (< median risk) and found that high risk (according to the scale developed in 1) have higher subway utilization during the pandemic despite similar ridership pre-pandemic. (3) The authors compare their risk index to COVID-19 mortality. They find that each unit increase in the risk index is associated with a 21% increase risk of mortality. Below, I list major (should be addressed) and minor (helpful for a reader if addressed) points roughly in the order of appearance in the manuscript. Importantly, I would like to note that I am not an expert in the Bayesian model used and suggest a statistical reviewer familiar with those methods should be included in the review process.

Major points:

- Please provide justification for the 10 covariates selected for the risk index. Given that this index forms the basis of the other analyses, justification for the inclusion (and implied exclusion) of covariates is warranted. The authors briefly touch on some of these covariates in the introduction but never explicitly make clear why they are included.
- While a single index has some appeal, it is not clear why this is preferable to (for example), only using proportion uninsured and median income — both of which can be readily interpretable to county-level officials and policymakers. How much *better* is the index?
- The results around mortality are thin. It is unclear they strengthen the authors' points. I would suggest either moving them to the Supplemental Materials and referencing them in the results about COVID cases or a more thorough discussion of the results.
- New York City is a fairly unique city within the US. The conclusion should include some mention of the generalizability of results (or lack thereof) to epidemics in other US cities (and increasingly in rural parts of the US).
- It is not clear what the application of the results are. That is, if the single index really is much better than univariate metrics, what are others to do with this information? As the authors (very appropriately) mention in their limitations, identifying a "high risk neighborhood" can lead to problematic stigmatization. This may be worth the risk if the index provides actionable information but it's not clear that case was made here.

Minor points:

- Referencs to the Supplemental Materials should be checked. For example, Supplemental Table 2 appears in the text before Supplemental Table 1.
- The subway ridership approach is interesting; however, it is not clear why this is more appropriate than actual measures of mobility (e.g., using Apple or Google Mobility Report, SafeGraph, etc.). UHF-level neighborhoods are much larger, resulting in a sample size of only 36 areas (18 in each group). Using other mobility data may provide a more convincing analysis.

Thank you for the opportunity to review this manuscript. I applaud the authors for taking on an important topic and providing code and data to replicate their analyses. I hope these comments are constructive and provide a clear path to an even stronger manuscript.

Reviewer #3 (Remarks to the Author):

This is a really fascinating paper that, while limited to a cross-sectional and ecologic study design, uses some unique methods to suggest that (1) the distribution of SES and housing-related variables are driving racial/ethnic differences in covid risk, and (2) changes in public transit use also play a role.

My suggestions are all fairly minor.

- The authors should describe the study design as ecological

- This description of inequalities in NYC seem to understate the inequalities:

"Hispanics/Latinx and Blacks are disproportionately impacted, representing 34% and 29% of the deaths, but 28% and 22% of the population, respectively"

I suggest looking at the data from the DOHMH website showing age-adjusted death rates (<https://www1.nyc.gov/site/doh/covid/covid-19-data.page>), which shows black/latinx rates are almost precisely double that of non-Hispanic whites

- In describing the relative weighting of components of the BWQS, it's important to note that the credible intervals overlap and your comparisons are referring to the point estimates

- I think it would be more beneficial to move the text explaining BWQS (including comparisons with PCA) that is currently under 'strengths' into the methods section, where I had expected to see it as I was reading.

- The authors should note that a key limitation is that cumulative incidence is based on confirmed positive cases, and $\Pr(\text{testing} \mid \text{infected})$ may vary by neighborhood. I am agnostic as to whether the author call the outcome measure "cumulative COVID-19 infection incidence", but it should also be clarified early on that it is defined by viral swab-confirmed infections.

Reviewer #4 (Remarks to the Author):

This is very timely research focused on an important question. In recent weeks, it's been clear that the COVID-19 – like almost any phenomenon – has a SES gradient in that it impacts the disadvantaged disproportionately more than anyone else. The authors to the extent I could understand – as there are multiple threads here – are trying to explain if neighborhood disparities (as captured thru an social

index) in COVID-19 infections and mortality on a SES index can be “explained” by these neighborhoods difficulty in practicing ‘social distancing’. I have some major concerns.

I still struggle with the discourse on COVID-19 on a compelling evidence that “social distancing” is a potent strategy. In some sense if this is translated to literally isolating each individual to a confined space that no one else can come inside and maintain that spatial fidelity then in theory yes. But this label is all over the place – both conceptually and it’s garden variety with re its practice. This then begs the very premise that’s being used to link SES disadvantage to infection/mortality.

I may have missed it but I could not find a clear test of SES index and social distancing relationship. I see that SES index is related to infection/mortality and the latter (writing and presentation could do with a clear flow or even a road map upfront as the authors here trying “explain” thru some “mediation” lens which is rather tricky) but not the first half of the relationship. So I can think of multiple DAGS underpinning these 3 primary phenomenon: social index, social distance and COVID.

A key data limitation is not being able to control for “pre-existing conditions”; while if I understand authors do a good job using that in their discussion but then that also takes away the “social distance” explanation. Additionally, even a proxy of that say thru “race” is missing.

In short, while I really like the question, the data limitations are critical to answer the question. Further, manuscripts needs to be organized much more clearly – perhaps with a visual and much greater justification is required on pinning this on social distance.

Reviewer #5 (Remarks to the Author):

Overview

In this study the authors use publicly available data to infer a metric for social disadvantage by using BWQS regression. They then compare this to COVID-19 cases adjusted for testing along with a measure or ability to socially isolate, subway ridership. This research proposes an interesting question and provides the framework for a link between social disadvantage and COVID-19 disparities. As it uses methods which have previously been used in environmental research (as far as I can tell), it would benefit from a deeper explanation of the methods and any checks for robustness that may have been conducted. Thank you for this interesting work!

Comments:

- This study would be strengthened greatly by a more in-depth explanation of the methods. Based on the references it seems that this methodology has primarily been used in environmental epidemiology, therefore, greater transparency in the analytic process is needed. A non-exhaustive list of consideration might include:

- o A deeper look into how the social variables were chosen including literature reviewed and the weight of subject matter expertise that was considered.
- o Which variables were considered then excluded and why?

- o Were sources other than the ACS considered?
- o Do you differentiate between “work from home” and “able to work from home”? The former might include no difference in individual risk while the latter likely signals a large disparity in ability to physically distance.
- o A description of the BWQS regression algorithm including any references to previous implementations of this method in social epidemiology.
- o Comparisons to other methods and why they may not be appropriate in this case.
- o Checks for robustness of the model.
- o Inclusion of other sources of mobility data (there is a plethora of mobile phone data that is freely available to researchers) which can further corroborate the conclusion.
- This discussion section could benefit from a paragraph suggesting how to best use this tool and your vision for the future. Among the social variables evaluated, essential worker status, insurance coverage and number of people per household stand out. Do you expect this to remain static? If there is another outbreak in New York City, would you expect the risk in each neighborhood to be similar? Could you perform a pre-hoc experiment by generating predictions of areas most likely to be affected in another city and compare it to outcomes? Given this information what sort of public health interventions do you imagine working well? If people are at higher risk because they are essential workers, uninsured or live in larger households, how do you imagine intervening on that?
- On the first line of page 6 you refer to figure 3 and a comparison figure in the supplement. It would be useful to have both maps as subfigures of figure 3 to ease comparison. It seems that all maps have been generated with different themes and aesthetic choices making direct comparison difficult.
- It would be worthwhile to perform the same analysis at the ZCTA level to demonstrate that the method and the results are robust. I imagine that individuals at a border may cross the UHF boundary to access other subways as well. In the worst-case scenario, I imagine that there must be some measure of catchment areas of each subway station which may be used to provide greater granularity. This would likely also be more operationally useful.
- Are there any locations where the risk metric doesn't perform well? If so could you provide more information as to why you think that might be the case and how to best account for it in implementation?

Dear reviewers,

Thank you for the opportunity to revise our manuscript, “Assessing capacity to social distance and neighborhood-level health disparities during the COVID-19 pandemic” for consideration in *Nature Communications*. We genuinely appreciate the detailed feedback in these five thoughtful reviews. We also appreciate the reviewers comments on the importance of the research question and the originality and contribution of our analyses including that “[we] address an important topic”, that ours is a “really fascinating paper” and “very timely research focused on an important question.” Most importantly, we appreciate that the reviewers recognize that, “this research proposes an interesting question and provides the framework for a link between social disadvantage and COVID-19 disparities.” We have responded to the reviewer feedback point-by-point below and have substantially improved the manuscript with detailed sensitivity analyses demonstrating robustness, a conceptual directed acyclic graph, and a critically revised discussion.

Among our many additions and modifications, we have updated the title to “Assessing capacity to social distance and neighborhood-level health disparities during the COVID-19 pandemic in New York City”, acknowledging that New York City is a particularly unique location in the United States, and worthy of its own emphasis given the experiences in the first wave of the COVID-19 pandemic. We have now incorporated several sensitivity analyses, in either the manuscript or code base, including: 1) assessing the impact of different priors for our negative binomial BWQS model, 2) creating our COVID-19 inequity index with Census tract-based summary measures, 3) comparing the BWQS model performance to adjusted regression models utilizing either principal components or just two selected top predictors, and 4) conducting our subway analysis with varied spatial scales and index cutoffs. Your comments also prompted us to conduct additional checks for robustness, which led us to change the parametrization of our adjustment for zip code-level testing from a linear adjustment to a more flexible cubic spline-based adjustment. We now provide additional visualizations for the subsequent robustness checks. Finally, we have changed the name of our BWQS neighborhood infection risk index to the COVID-19 inequity index. We felt this was necessary to recenter the emphasis on the complex and multidimensional social and infection disparities rather than potentially stigmatizing neighborhoods based on infection levels.

We believe this work is as relevant now as when we originally submitted it in early June. The historical experience of New Yorkers during this time was unprecedented, and helped to sound the alarm of COVID-related disparities for the rest of the country. And those COVID-19 disparities are persisting in NYC according to the NYC Department of Health data. We believe that our unique analysis of the first wave of the pandemic is instructive to how we think about pandemic preparedness in the future, specifically given that we so clearly tie COVID-19 disparities to long-standing social inequities. Finally, thanks in part to your helpful feedback, we believe that this is a methodologically rigorous and novel approach to addressing these

questions and remains unique in the level of transparency and reproducibility that we offer through our use of publicly available data with posted code.

REVIEWER COMMENTS

Reviewer #1 (Remarks to the Author):

This paper examines the associations between a social disadvantage index and COVID-19 infections/mortalities and also explores how subway ridership in different neighborhoods changed around the implementation of the 'NYS on PAUSE' social distancing policy. My comments on this paper are below.

1. I am not exactly sure how the social distance/subway ridership analysis fits into the broader narrative of this study. Is it meant to illustrate one possible mechanism linking disadvantage and COVID? If so, please justify why subway ridership is a good variable for representing the capacity to social distance. It is not clear - either from the paper or from my intuition - why zip codes with higher subway ridership would have lower levels of social distancing. Riding the subway is only one of many possible exposure settings for COVID and only one of many possible ways to (not) practice social distancing. Has this variable been validated in other research vis a vis more direct measures of social distancing (e.g., movement data of cell phones or similar)?

Authors' response: Thank you for this important comment. We have added additional text to the results to address this issue, quoted below. We have also referenced two other manuscripts in the discussion that tie mobility (cell phone data and subway) to ZIP code-level infection rates, which come to similar conclusions, references below. Finally, we have also now included Supplemental Figure 9, which shows Google Mobility data in comparison to subway utilization. The figure demonstrates that the trend is the same between the two measures, but the magnitude differs, likely because Google Mobility also captures bus and regional rail ridership, although it is only available at the much larger county (borough) level.

"We used area-level subway ridership as a proxy for the capacity to socially distance, and to examine differences in ridership by our measure of neighborhood disadvantage, thus representing a possible mechanism between neighborhood disadvantage and infections. Given that neighborhood disadvantage is spatially heterogeneous, associated ridership differences may indicate places where individuals were less able to socially distance due to being an essential worker or not having access to a car and living in denser housing conditions."

2. What is the rationale for creating a social disadvantage index and why is this index better - for representing different types of neighborhood environments and for informing policy - than simply analyzing a smaller set of variables (not transformed into an index)? I understand that socioeconomic variables are often highly correlated and so multicollinearity is a concern, but is multicollinearity actually an issue in this study? Are there theoretical or policy-oriented reasons

as to why a single index does a better job capturing neighborhood characteristics than multiple variables?

Authors' response: Thank you for raising these important points. We have now included a correlation plot in Supplemental Figure 1 that demonstrates the correlation between the variables, which ranges from -0.15 to 0.61. We have also included additional text in the discussion about the potential contributions of an index. In summary, we think that an index can be used to target broad interventions, while the weights contributing to an individual area's index can be used to tailor specific interventions to the main drivers of infection.

“The appeal of an index is both summative and contextual. The composite index can be used to direct outreach and testing (such as mobile testing units²⁶) regardless of the components most driving the index value in each geographic unit. Also, index weights can be used to inform the types of interventions targeted to specific neighborhoods. In NYC, COVID-19 positive residents could stay in hotels for two weeks until they were no longer contagious²⁷, but it is unclear if this was a targeted intervention. Our index helps contextualize the unique combination of social conditions per ZCTA that relate to ZCTA-level infections for targeting interventions (e.g. in ZCTAs with higher average number of people per household, then rapid testing could be paired with outreach for hotel isolation).”

3. One additional limitation of this study that deserves some discussion is the use of zip code-level data. I can think of two issues in particular. One, zip codes are quite large and are likely internally heterogeneous with respect to the COVID variables and disadvantage. Two, because of this high internal heterogeneity and because zip codes are not the same units of analysis used by health policy professionals, they may not be suitable for designing/implementing policies.

Authors' response: This is an important and thoughtful point. Given that New York City, and many other state and local governments, are only releasing infection data at the ZIP code level (in order to preserve privacy), this is indeed a challenge that most other research groups have had. This has now been mentioned in our limitations, see below. In addition to including the variation in size in the ZCTAs within NYC (2018 ACS populations ranging from 3,028 to 112,425), we also used your comment as an opportunity to innovate since ZCTA-level mean summaries can hide meaningful social disparities (a wealthy neighborhood next to a poorer neighborhood is not the same as two middle-income neighborhoods). We have now included novel sensitivity analyses whereby we assess the results of two tract-level social indicators as our independent variables: 1) a weighted-quantile median and 2) a weighted quantile third quartile representing those with greater social disadvantage within the ZCTA (see the Methods section, “Robustness of ZCTA-level measures” for additional detail). We found that the tract-level analyses actually performed better (lower WAIC) than the ZCTA-level analysis. However, the improvement was small, and we felt that maintaining the same administrative units for the exposure and outcome was easier for interpretation, particularly as the relation between the smaller tracts, which are not necessarily aligned well with ZCTAs, requires a complex

reweighting based on HUD-supplied crosswalk tables. Nonetheless, it points to the importance of finer level analyses when possible.

“Seventh, this study uses ZCTAs as our administrative unit of analysis. There is likely greater demographic heterogeneity in some ZCTAs compared to others, and few health-related decisions are made on this level. However, the NYC and NYS DOH have only released data at this scale, thus limiting our ability to examine relationships at varying spatial scales. We assessed the sensitivity of our results to using tract-level data to estimate our ZCTA-level exposure metrics, and found relatively consistent results.”

4. Did the authors complete any sensitivity analyses for the prior distributions in the BWQS model? I am particularly interested in if/how the model results may change when the Inverse Gamma(0.01, 0.01) prior for the overdispersion parameter is modified.

Authors’ response: While we have chosen minimally informative priors, we have now also conducted a sensitivity analysis in which we use a half-Cauchy(0, 3) distribution for the overdispersion parameter instead of the Inverse Gamma(0.01, 0.01) in our main model. We found that this had no meaningful difference in our effect estimate (1.077191 versus 1.077389) or the associated credible interval (1.065459, 1.089427 versus 1.065098, 1.089121) for *beta1* (the BWQS term) and the WAIC differed by less than 0.05 indicating that the model was not sensitive to these uninformative priors. This implementation using the alternate prior distribution is now included in our supplementary materials and in our public code in the sensitivity analysis section.

5. Are the results presented in Figure 5 sensitive to the choice of two groups (above median BWQS index vs. below median BWQS)? Supp Fig 2 breaks down the racial/ethnic composition in 3 zip code groups, so why does Fig 5 focus on only 2 groups? Is it possible that Fig 5 may show a different pattern if the ridership curves are fit to 3 or 4 groups?

Authors’ response: Thank you for your question. We have now included Supplemental Figure 7, which models the subway ridership for three instead of two risk groups at the UHF level. As prompted by another reviewer comment, we have done the same thing at the ZIP code level as well (Supplemental Figure 8). We have also modeled the data continuously in linear models with cubic splines (not shown). Each modeling approach has demonstrated the same trends and consistent results.

6. Under 'Capacity to Social Distance' in the Methods section, the authors note that they want to 'assess the degree to which infections were explained by inability to socially isolate/distance'. But the analysis that addresses this statement focuses on BWQS index values and not COVID infection rates, so there appears to be some inconsistency in the language here.

Authors’ response: Thank you for your comment. We agree and have updated the language accordingly.

“Our BWQS model uses cross-sectional data to create an infection risk index, but we wanted to assess the degree to which those differences in infections were explained

longitudinally by inability to socially isolate/distance. We could not assess this directly, so we chose to look at subway ridership in relation to neighborhood infection risk.”

7. It would be helpful if the authors could provide formal notation of the BWQS model in the Supplementary Material.

Authors’ response: We have now included formal notation and explanation of the BWQS model in our Supplemental Methods section. In addition, the BWQS package is now publicly available at <https://github.com/ElenaColicino/bwqs>.

8. Please include a map of the model residuals in the Supplementary Materials to support the claim that there was no residual spatial autocorrelation. A visual inspection of the model residuals is just as informative as the Moran’s I diagnostic.

Authors’ response: Thank you for this important point. We have now included a spatial representation of the model residuals in Supplemental Figure 11. In addition, upon conducting sensitivity analyses of our results, we decided to modify our nearest neighbor spatial averaging strategy from 4 to 6, which resulted in a lower Moran’s I (0.07 to 0.05) with a higher p value (0.06 to 0.08) and a slightly attenuated point estimate for the BWQS index (1.21 to 1.2). We have updated the results in the manuscript.

9. Add a scale bar to the maps. This will help readers who are unfamiliar with NYC zip codes to interpret the size of the areas analyzed.

Authors’ response: We have now added a scale bar to the maps. We have also included in our text the range of population (3,028 to 112,425 from the 2018 ACS) for the 177 modified ZCTAs used by NYC DOHMH.

10. Please define the acronym UHF when it is first introduced in the paper. I think that this may be an issue with my version of the paper as the data section is after the discussion section.

Authors’ response: We have now defined UHF upon its introduction in the results section.

11. While it is hard to keep up with the new papers in this field, the results presented here are original to the best of my knowledge. Regardless, I think that it would be prudent to add some discussion as to how this study is similar/different from other studies looking at zip code COVID data. Specifically, (a) Chen and Krieger 'Revealing the unequal burden...', (b) DiMaggio et al. 'Blacks/African Americans are 5 Times More Likely to Develop COVID-19', (c) Borjas 'Demographic Determinants of Testing Incidence', and (d) Schmitt-Groh?? et al. 'COVID-19 Testing Inequality in New York City'

Authors’ response: Thank you for prompting us to situate our findings in the other published and pre-print manuscripts on this topic. We have added to the discussion section reviewing the results of other studies, including all those you mentioned above and some others we found. Please find the text below.

“A growing body of literature is examining the greater impact of COVID-19 based on measures of neighborhood structural disadvantage. As some have noted, COVID-19 is not creating new health disparities, but exacerbating those that already exist³. A recent investigation found that county and ZCTA area-based socioeconomic measures, specifically using crowding, percent POC, and a measure of racialized economic segregation, were useful in identifying higher COVID-19 infections and mortality in Illinois and New York¹⁹. Work on COVID-19 mortality in Massachusetts has found excess death rates for areas of higher poverty, crowding, proportion POC, and racialized economic segregation²⁰. There has been some concern that neighborhood disadvantage and case positivity associations could be confounded by access to testing, with advantaged communities having more access than lower²¹; however, Schmitt-Grohe et al. found no evidence of testing inequalities by income, and a small negative association when also including proportion Black²². An analysis examining spatial patterns of infections found that ZCTAs with high proportions of Black residents and residents with chronic obstructive pulmonary diseases (COPD) were the most likely to experience the highest COVID-19 infection rates²². The utility of these findings are opaque, given that there is no evidence, to our knowledge, that COPD increases the likelihood of COVID-19 exposure and infection. We know of two other studies that have tied neighborhood-level disadvantage mobility (via cellphone data)²¹ and subway utilization²², and mobility data with COVID-19 infections^{21,23}. Researchers have begun to identify counties that are particularly susceptible to severe COVID-19 outcomes using a combination of biological, demographic, and socioeconomic variables²³. They identified counties with high population density, low rates of health insurance, and high poverty as particularly at risk. However, a stated limitation of this work is that many of these variables are interrelated.”

Reviewer #2 (Remarks to the Author):

In this paper, the authors address an important topic and seek to assess the role of neighborhood-level structural disadvantage on observed racial/ethnic inequalities in COVID-19 cases and deaths as well as the ability for disadvantaged groups to socially distance. The paper presents the results of three main analyses. (1) A risk index is created using Bayesian weighted quantile sums regression on 10 neighborhood-level covariates and positive tests. The authors find that a one unit increase in this risk index is associated with a 110% increase in infections per capita. (2) Using subway ridership data, the authors compared the high risk neighborhoods (> median risk) to low risk neighborhoods (< median risk) and found that high risk (according to the scale developed in 1) have higher subway utilization during the pandemic despite similar ridership pre-pandemic. (3) The authors compare their risk index to COVID-19 mortality. They find that each unit increase in the risk index is associated with a 21% increase risk of mortality. Below, I list major (should be addressed) and minor (helpful for a reader if addressed) points roughly in the order of appearance in the manuscript. Importantly, I would like to note that I am not an expert in the Bayesian model used and suggest a statistical reviewer familiar with those methods should be included in the review process.

Major points:

1) Please provide justification for the 10 covariates selected for the risk index. Given that this index forms the basis of the other analyses, justification for the inclusion (and implied exclusion) of covariates is warranted. The authors briefly touch on some of these covariates in the introduction but never explicitly make clear why they are included.

Authors' response: Thank you for this important point, which was also raised by another reviewer. We have used this as an opportunity to expand on our approach in the methods section. Specifically we have included the following text:

“Candidate demographic variables for the BWQS infection risk index from the ACS included income, rent, households using supplemental nutrition assistance program (SNAP) benefits, poverty, health insurance, unemployment, and industries of employment. Derived variables from ACS data included average rent burden and household size. Non-ACS variables included the population density (persons per square foot of the ZCTA) and residential population density (persons per cubic foot of ZCTA residential volume). Our choice of variables is largely influenced by the theoretical framework from Acevedo-Garcia¹² and our understanding of the employment sectors deemed essential workers, and therefore less able to stay home during a time of social distancing². Our measures of population density attempt to capture (in)ability to physically distance within the home and otherwise dense housing conditions such as apartment buildings versus single family homes, and therefore higher risk of contact with infected individuals. Finally, SNAP benefits and measures of food access are included to indicate further travel from home for basic necessities and/or less opportunity to amass food reserves to reduce overall frequency of shopping. We restricted candidate variables to those that would be realistically publicly available or accessible in other parts of the United States.”

2) While a single index has some appeal, it is not clear why this is preferable to (for example), only using proportion uninsured and median income ??? both of which can be readily interpretable to county-level officials and policymakers. How much *better* is the index?

Authors' response: We have now added comparisons of the BWQS regression to 1) a negative binomial model with proportion essential workers and median income based on the results of another paper: Sy et al., 2020 <https://www.medrxiv.org/content/10.1101/2020.05.28.20115949v1> and 2) a regression using the largest principal component following PCA. We found that the BWQS regression results in the smallest root mean squared error and the largest Kendall's tau (ranking ZCTA-level infection risks) when compared to these alternate modeling approaches (Supplemental Table 4). Furthermore, we point out that the BWQS provides variable importance weights to guide the prioritization of public health resources. Thank you for prompting these additional analyses.

3) The results around mortality are thin. It is unclear they strengthen the authors' points. I would suggest either moving them to the Supplemental Materials and referencing them in the results about COVID cases or a more thorough discussion of the results.

Authors' response: While we focus on constructing an infection risk index based on the primary association of pre-existing social determinants with neighborhood level infection rates, we believe it is an important consequence that this infection risk index derived from social conditions also has a strong association with community-level COVID-19 mortality. While disparities in COVID-19 have been noted in both infection and mortality, they are not always linked in the literature. We believe that our mortality analysis is important because it demonstrates a pathway through which social determinants that relate to infection can also explain neighborhood patterns in mortality. Below is additional context we have provided in the results section:

“There were 16,289 COVID-related deaths across 177 ZCTAs by May 23, 2020. NYC DOHMH surveillance data shows that race/ethnic disparities are greater for COVID-19 mortality than for SARS-Cov-2 infections. Therefore, we wanted to assess whether our index that captures neighborhood structural disadvantage in relation to infection was also related to ZCTA-level mortality.”

4) New York City is a fairly unique city within the US. The conclusion should include some mention of the generalizability of results (or lack thereof) to epidemics in other US cities (and increasingly in rural parts of the US).

Authors' response: We agree. Not only have we added this to our conclusion (see below), but we have also changed the title of the manuscript to: *Assessing capacity to social distance and neighborhood-level health disparities during the COVID-19 pandemic in New York City*. We believe this will accurately convey our scope and focus to future readers.

“This is an important area of investigation given the large toll that COVID-19 has had, and will likely continue to have unless action is taken, on disadvantaged communities of color in NYC and elsewhere. Future work should assess the generalizability of these results in other parts of the country, new waves of the pandemic, or if our approach can be adapted to different contexts, potentially yielding different social variables that are associated with increased COVID-19 inequities.”

5) It is not clear what the application of the results are. That is, if the single index really is much better than univariate metrics, what are others to do with this information? As the authors (very appropriately) mention in their limitations, identifying a "high risk neighborhood" can lead to problematic stigmatization. This may be worth the risk if the index provides actionable information but it's not clear that case was made here.

Authors' response: Thank you for raising these important points. We have included additional text in the discussion about the potential contributions of an index. In summary, we think that an index can be used to target broad interventions, while the weights contributing to an individual area's index can be used to tailor specific interventions to the main drivers of infection. We have added the text below to the discussion section of the manuscript. Further, we have decided to change the name of the index from the 'neighborhood infection risk index' to the 'COVID-19 inequity index'. Your comment made us realize that the original name was emphasizing risk

based on neighborhoods, which simply worked against our stated desire not to stigmatize areas. Instead, we feel that the new name recenters the emphasis on the role of social inequities that drive infection risks, which is more in line with our intended goal and the literature as a whole. We added an additional paragraph emphasizing the importance of intervening on inequities rather than “neutral” interventions that ultimately result in inequitable outcomes. Thank you again for prompting these changes, clarifications, and improvements.

“The appeal of an index is both summative and contextual. The composite index can be used to direct outreach and testing (such as mobile testing units²⁶) regardless of the components most driving the index value in each geographic unit. Also, index weights can be used to inform the types of interventions targeted to specific neighborhoods. In NYC, COVID-19 positive residents could stay in hotels for two weeks until they were no longer contagious²⁷, but it is unclear if this was a targeted intervention. Our index helps contextualize the unique combination of social conditions per ZCTA that relate to ZCTA-level infections for targeting interventions (e.g. in ZCTAs with higher average number of people per household, then rapid testing could be paired with outreach for hotel isolation).”

“Our work focused on the social conditions that allowed for uneven exposure to SARS-CoV-2, and higher infections, for disadvantaged communities in the spring of 2020. Since then, this situation has been compounded by national interventions that do not appear to be yielding equitable outcomes. For example, the federal Paycheck Protection Program is largely not accessible to minority-owned businesses³⁴ and bias has been found in the disbursement of Coronavirus Aid, Relief, and Economic Security act funding, with hospitals in predominantly Black communities receiving fewer dollars on average³⁵. These examples underscore an important reality; addressing health inequities requires explicit identification of disparities, their derivation, and creating interventions designed for equitable outcomes. This is essentially the strategy behind the National Academies of Sciences, Engineering, and Medicine’s Consensus Report on the allocation of COVID-19 vaccinations, which proposes targeted distribution based on measures of social vulnerability³⁶. Consequently, we believe that tailored health and social equity initiatives represent an important path forward for COVID-19 pandemic response and planning.”

Minor points:

6) References to the Supplemental Materials should be checked. For example, Supplemental Table 2 appears in the text before Supplemental Table 1.

Authors’ Response: Thank you for identifying this mistake. We have updated the references to the figures and tables accordingly.

7) The subway ridership approach is interesting; however, it is not clear why this is more appropriate than actual measures of mobility (e.g., using Apple or Google Mobility Report, SafeGraph, etc.). UHF-level neighborhoods are much larger, resulting in a sample size of only 36 areas (18 in each group). Using other mobility data may provide a more convincing analysis.

Authors' Response: Thank you for this suggestion. We have addressed this issue in two ways. First, we have now co-plotted the Google Mobility trends data per NYC borough (county) with our subway trends data (Supplemental Figure 9). We find that the trends are largely the same, but different in magnitude, which is intuitive because the Google Mobility data captures all public transit utilization (subway, buses, and regional rail lines), although it is limited due to its availability only at the borough (county) level. Second, we have now conducted our subway utilization analysis at the ZCTA level. This can be found in Supplementary Figure 8. We found the same trends as our UHF level analysis.

Thank you for the opportunity to review this manuscript. I applaud the authors for taking on an important topic and providing code and data to replicate their analyses. I hope these comments are constructive and provide a clear path to an even stronger manuscript.

Reviewer #3 (Remarks to the Author):

This is a really fascinating paper that, while limited to a cross-sectional and ecologic study design, uses some unique methods to suggest that (1) the distribution of SES and housing-related variables are driving racial/ethnic differences in covid risk, and (2) changes in public transit use also play a role.

My suggestions are all fairly minor.

1) The authors should describe the study design as ecological

Authors' response: Thank you for this important clarification; we have now added the sentences below to the introduction and discussion sections to indicate the ecological/population-based nature of our study design.

***Introduction:** "In this population-level study, we use socioeconomic data on neighborhood characteristics to understand differences in infection incidence between neighborhoods, as we quantify the relative contribution of these measures of social disadvantage and if a proxy of social isolation, NYC subway utilization, helps us to understand these differences."*

***Discussion:** "We conducted an ecological study using publicly-available data to identify the role of neighborhood social disadvantage on cumulative COVID-19 infections and COVID-19-related mortality."*

2) This description of inequalities in NYC seem to understate the inequalities:

"Hispanics/Latinx and Blacks are disproportionately impacted, representing 34% and 29% of the deaths, but 28% and 22% of the population, respectively"

I suggest looking at the data from the DOHMH website showing age-adjusted death rates (<https://www1.nyc.gov/site/doh/covid/covid-19-data.page>), which shows black/latinx rates are almost precisely double that of non-Hispanic whites

Authors' response: Thank you for this important point. We have now replaced the text with crude rates as drawn from the New York City Department of Health and Mental Hygiene's COVID-19 dashboard.

"In New York City (NYC), Hispanics/Latinx and Blacks are disproportionately impacted, with mortality rates of 264 and 249 per 100,000 respectively, compared to 124 for Whites."

3) In describing the relative weighting of components of the BWQS, it's important to note that the credible intervals overlap and your comparisons are referring to the point estimates

Authors' Response: Thank you for your comment. Yes, this is correct, and we have updated the text to reflect our interpretation of the effect estimates and not the overlapping credible intervals.

"While all included variables contributed to this composite, they do not all contribute equally (Figure 2). We interpret the point estimates of the weight assigned to the social variables, noting that the credible intervals of the variables overlap with one another."

Furthermore, because the weights are constrained to sum to 1 (and thus magnitudes are interrelated) we also summarize uncertainty as the proportion of draws from the posterior of the distribution for the weights in which each social variable has the largest weight (Supplemental Table 3).

4) I think it would be more beneficial to move the text explaining BWQS (including comparisons with PCA) that is currently under 'strengths' into the methods section, where I had expected to see it as I was reading.

Authors' Response: Thank you for this suggestion. We have now updated the text, and have included the comparison to the PCA in the methods section. Specifically we state:

"Researchers have often addressed the multicollinearity of social determinants with the use of dimensionality reduction techniques such as principal components analysis (PCA) in the case of the neighborhood deprivation index. However, traditional PCA only considers correlations between SES variables, whereas a supervised method captures features most relevant for the outcome. To address these shortcomings, we developed a weighted combination of socioeconomic variables to explain the cumulative number of COVID-19 cases per ZCTA using Bayesian weighted quantile sums regression (BWQS)."

5) The authors should note that a key limitation is that cumulative incidence is based on confirmed positive cases, and $\Pr(\text{testing} \mid \text{infected})$ may vary by neighborhood. I am agnostic as

to whether the author call the outcome measure "cumulative COVID-19 infection incidence", but it should also be clarified early on that it is defined by viral swab-confirmed infections.

Authors' Response: This is an important clarification, which we have reflected at the beginning of the results section.

"We wanted to identify any association between a neighborhood social disadvantage composite index and cumulative COVID-19 viral swab-confirmed infection incidence."

We have also included a more flexible parameterization for the association of testing and infection using a natural spline with 3 degrees of freedom (see response to Reviewer 5 Question 7 below and Supplemental Figure 2) to allow this relationship to vary.

Reviewer #4 (Remarks to the Author):

This is very timely research focused on an important question. In recent weeks, it's been clear that the COVID-19 like almost any phenomenon has a SES gradient in that it impacts the disadvantaged disproportionately more than anyone else. The authors to the extent I could understand as there are multiple threads here are trying to explain if neighborhood disparities (as captured thru an social index) in COVID-19 infections and mortality on a SES index can be explained by these neighborhoods difficulty in practicing social distancing. I have some major concerns.

1) I still struggle with the discourse on COVID-19 on a compelling evidence that social distancing is a potent strategy. In some sense if this is translated to literally isolating each individual to a confined space that no one else can come inside and maintain that spatial fidelity then in theory yes. But this label is all over the place both conceptually and it's garden variety with re its practice. This then begs the very premise that's being used to link SES disadvantage to infection/mortality.

Authors' response: Thank you for your comment, as this is an important issue to unpack. Conceptually, our argument is that neighborhood-level factors preceding the pandemic are associated with people's ability to mitigate their exposures to COVID-19 infected people. We should note that New York City guidance acknowledges your point that spatial fidelity within households is unlikely, and therefore social/physical distancing is suggested between households. As we have demonstrated in our analysis, there is community-level heterogeneity in factors that increase the likelihood of persistent social contact between households during NYS on PAUSE, such as the estimated proportion of essential workers and measures of population density. As such, residents of various communities simply have a different menu of options in order to mitigate their own household's risk of infection.

2) I may have missed it but I could not find a clear test of SES index and social distancing relationship. I see that SES index is related to infection/mortality and the latter (writing and presentation could do with a clear flow or even a road map upfront as the authors here trying

???explain??? thru some ???mediation??? lens which is rather tricky) but not the first half of the relationship. So I can think of multiple DAGS underpinning these 3 primary phenomenon: social index, social distance and COVID.

Authors' Response: We have added a conceptual DAG (see Supplemental Figure 12) to help explain the relationships that we are investigating. In particular, we show that the infection risk index based on social conditions is related to the capacity to social distancing, but that the subway utilization that we derive is a proxy for this capacity and is not implied as the source/route for infections. Instead, it is its own manifestation of social distancing because people who are riding the subway during this time are not only leaving the household, and they are likely interacting with/in proximity to others outside of their household either on the subway or their place of work. In drawing the DAG, we realized that area level chronic disease or age distributions would serve as effect modifiers rather than confounders in our mortality analysis. Consequently, we felt that adjustment was inappropriate and removed its adjustment from the analysis.

3) A key data limitation is not being able to control for ???pre-existing conditions???; while if I understand authors do a good job using that in their discussion but then that also takes away the ???social distance??? explanation. Additionally, even a proxy of that say thru ???race??? is missing.

Authors' Response: The focus of our analysis is how social conditions relate to the capacity to avoid infection in the earliest months of the outbreak and implementation of the NYS on Pause policies. While we agree that pre-existing conditions and older age play an important role in the likelihood of severe disease (or death) after infection, we disagree that differences in the prevalence of these between neighborhoods (which are also driven by the kinds of social factors we relate to infection risk) are likely explanations for differences in infections. The relationships that we investigate are now displayed visually in a DAG (Supplemental Figure 12).

4) In short, while I really like the question, the data limitations are critical to answer the question. Further, manuscripts needs to be organized much more clearly ??? perhaps with a visual and much greater justification is required on pinning this on social distance.

Authors' Response: We have substantively reorganized and added additional explanation of the relationships that we investigate with our ecological modeling. While we agree there are important data limitations, we believe that our detailed modeling elucidates relationships explained by preexisting social conditions with major implications for understanding disparities in COVID-19 infections and mortality during the Spring of 2020 in New York City.

Reviewer #5 (Remarks to the Author):

In this study the authors use publicly available data to infer a metric for social disadvantage by using BWQS regression. They then compare this to COVID-19 cases adjusted for testing along with a measure or ability to socially isolate, subway ridership. This research proposes an interesting question and provides the framework for a link between social disadvantage and

COVID-19 disparities. As it uses methods which have previously been used in environmental research (as far as I can tell), it would benefit from a deeper explanation of the methods and any checks for robustness that may have been conducted. Thank you for this interesting work!

Comments:

- This study would be strengthened greatly by a more in-depth explanation of the methods. Based on the references it seems that this methodology has primarily been used in environmental epidemiology, therefore, greater transparency in the analytic process is needed. A non-exhaustive list of consideration might include:

- 1) A deeper look into how the social variables were chosen including literature reviewed and the weight of subject matter expertise that was considered.
- 2) Which variables were considered then excluded and why?
- 3) Were sources other than the ACS considered?

Authors' response: Thank you for prompting these important clarifications. First, we compiled literate programming html report of all code with all generated output, now available at https://justlab.github.io/COVID_19_admin_disparities. We hope this satisfies concerns on transparency of our analytic process since it reflects each analytical choice made and the resulting outputs. Further, we have added additional text to the methods section, which we believe will clarify comments 1-3.

“Candidate demographic variables for the BWQS infection risk index from the ACS included average household size, income, rent, households using supplemental nutrition assistance program (SNAP) benefits, poverty, health insurance, unemployment, and industries of employment. Derived variables from ACS data included average rent burden and household size. Non-ACS variables included the population density (persons per square foot of the ZCTA) and residential population density (persons per cubic foot of ZCTA residential volume). Our choice of variables is largely influenced by the theoretical framework from Acevedo-Garcia and our understanding of the employment sectors deemed essential workers, and therefore less able to stay home during a time of social distancing. Our measures of population density attempt to capture (in)ability to physically distance within the home and otherwise dense housing conditions such as apartment buildings versus single family homes, and therefore higher risk of contact with infected individuals. Finally, SNAP benefits and measures of food access are included to indicate further travel from home for basic necessities and/or less opportunity to amass food reserves to reduce overall frequency of shopping. We restricted candidate variables to those that would be realistically publicly available or accessible in other parts of the United States.”

- 4) Do you differentiate between ???work from home??? and ???able to work from home????? The former might include no difference in individual risk while the latter likely signals a large disparity in ability to physically distance.

Authors' response: Thank you for this point of clarification. In our BWQS model, we do indeed differentiate between these two types of work from home. 1/work from home is based on a Census variable B08301_021, which is the number of individuals in an area that report regularly working from home in the 2018 ACS. On the other hand, our essential workers variables are based on our approximations of industries that were deemed essential versus not. Among those results, essential workers who drove personal vehicles had the highest weights in the BWQS model, followed by 1/work from home. We agree that there are many different potential explanations for these differences.

5) A description of the BWQS regression algorithm including any references to previous implementations of this method in social epidemiology.

Authors' Response: Thank you -- this is an important clarification. We have also added information to the supplemental on the details of the method, including formal notations. Below is also text we included in the discussion.

***Discussion:** "Sixth, this study is the first time the BWQS method has been applied to social/infectious disease epidemiology, although it has been successfully used in environmental epidemiology, which has similar issues of correlated multi-dimensional exposures."*

6) Comparisons to other methods and why they may not be appropriate in this case.

Authors' response: Please also see our reply to Reviewer 2 Question 2. We have now added comparisons of the BWQS regression to 1) a negative binomial model with proportion uninsured and median income and 2) a regression using the first principal component following PCA. We found that the BWQS regression results in the smallest mean absolute error and the largest Kendall's tau of ZCTAs infection rates versus predictions, when compared to these alternate modeling approaches (Supplemental Table 4). Thank you for prompting these additional analyses.

7) Checks for robustness of the model.

Authors' response: We have added a large number of sensitivity analyses and made changes to improve the robustness of our model and interpretation.

We have added to our listed strengths of the BWQS model that the quantile transform of the input variables makes it less sensitive to extreme values.

We have also added a visualization of the quantiled residuals from our BWQS model (Supplemental Figure 3) as well as several diagnostics for the parameter estimates in our supplement (Supplemental Table 2) which do not deviate from expectations. Similarly, we have added a map of the residuals from our spatial model of mortality (Supplemental Figure 11).

As we replied to Review 1 Comment 4, we have also conducted a sensitivity analysis for an alternate parameterization of the prior for overdispersion and found that it did not alter the effect estimate or credible interval for our BWQS term.

To assess the stability of our social variable weights, which may be useful for prioritizing public health interventions, we now report the proportion of posterior simulations in which each yielded the highest weight (Supplemental Table 3).

In response to Reviewer #1's concern that heterogeneity within ZCTAs could mask the risks associated with social disparities, we have used substantially finer tract-level ACS data and a HUD cross-walk (ZCTA-to-tract apportionment of households) table to estimate the within-ZCTA empirical cumulative distribution function of our social variables for sensitivity analyses. We have shown that our results are slightly improved in WAIC and Bayesian R^2 but have substantively the same effect estimates when we use the median or the 3rd quartile of the social disadvantage within each ZCTA as an alternative exposure metric for constructing our ZCTA-level BWQS index.

Although previous publications on COVID-19 positive cases in New York City adjust linearly for the number of tests conducted, we have now relaxed this assumption. As shown by the red line in this figure, the exponential shape of a linear term (in an unadjusted negative binomial regression) greatly exceeds the distribution of the data at the upper end of the testing ratio. We instead now utilize a parsimonious natural spline (using 3 degrees of freedom) instead of the linear parameterization and show that this smoother (in blue) better follows the data and is also consistent with the shape of a penalized spline from a generalized additive model (in green).

8) Inclusion of other sources of mobility data (there is a plethora of mobile phone data that is freely available to researchers) which can further corroborate the conclusion.

Authors' response: Thank you for this important comment. We used turnstile entrances and exits from MTA as a measure of mobility, but there are other potential data sources. Google

(<https://www.google.com/covid19/mobility/>) and Apple (<https://covid19.apple.com/mobility>) both provide county-level mobility reports, and have a transit category that we would expect to be most similar to our turnstile data. Safegraph also provides social distancing metrics at Census block group level to registered researchers (<https://www.safegraph.com/covid-19-data-consortium>).

Supplemental figure 9 compares turnstile counts from MTA to transit location visits from Google. Both are compared to earlier pre-pandemic baseline values. Turnstile counts are relative to the median daily count on the same day of the week in the same month throughout 2015-2019. Google transit location visits are relative to the median value between Jan 3 – Feb 6, 2020 for the same day of the week. MTA and Google data largely agree in the direction of trend, with Google data showing a smaller magnitude of reduction in transit utilization.

Staten Island has a much smaller rail network than the other boroughs: a single rail line with 21 stops compared to the over 400 stops spread across the other four boroughs. This helps explain the higher variability of Staten Island MTA data. It may also explain the greater difference between Staten Island MTA data and Google mobility estimates, as Google's transit category includes bus and ferry terminals, which may account for a higher share of all transit use on Staten Island than in the other boroughs.

One advantage of using turnstile data is its high spatial resolution. Turnstile counts are provided at the station level, which we can aggregate to UHF neighborhoods to estimate social distancing at much smaller regions than the counties used by Google and Apple. It is also uncertain how long the reports generated by these companies will remain available, whereas the data made available by MTA is routine, and not due to the COVID-19 pandemic.

In our exploratory analysis of Safegraph data, we had concerns whether the estimated home locations for individuals were correct for our period of interest. Some people left the NYC region during the peak of the pandemic, but the Safegraph algorithm determines home location based on a 6-week window (<https://docs.safegraph.com/docs/social-distancing-metrics>), and we do not expect it to immediately reflect the movement of people out of NYC. The algorithms used by Safegraph, Google and Apple to estimate mobility are proprietary, and the original data sources for all of these mobility alternatives (the individual, identifiable mobility data) are private. The public MTA turnstile data is also easier to interpret because it has not been through a classification algorithm prior to our use of it. That said, other manuscripts have used mobile phone data and subway data, and have come to similar conclusions about neighborhood disadvantage and mobility. We have placed this information in the discussion for additional context for the reader.

“To our knowledge, two other studies have tied neighborhood-level disadvantage to mobility (via cellphone data)²⁰ and subway utilization²², and mobility data with COVID-19 infections^{20,22}.”

9) This discussion section could benefit from a paragraph suggesting how to best use this tool and your vision for the future. Among the social variables evaluated, essential worker status, insurance coverage and number of people per household stand out. Do you expect this to remain static? If there is another outbreak in New York City, would you expect the risk in each neighborhood to be similar? Could you perform a pre-hoc experiment by generating predictions

of areas most likely to be affected in another city and compare it to outcomes? Given this information what sort of public health interventions do you imagine working well? If people are at higher risk because they are essential workers, uninsured or live in larger households, how do you imagine intervening on that?

Authors' response: Thank you for raising these important points. We have included additional text in the discussion about the potential contributions of an index and its potential utility for public health interventions. In summary, we think that an index can be used to target broad interventions, while the weights contributing to an individual area's index can be used to tailor specific interventions to the main drivers of infection. Most importantly, we provide an quantitative ecological set of results that show that pre-existing social disparities are strongly related to ZCTA-level differences in infections and also capture race/ethnic disparities and relate to mortality in the spring 2020 wave of COVID-19 in NYC. We have added the text below to the discussion section of the manuscript.

"The appeal of an index is both summative and contextual. The composite index can be used to direct outreach and testing (such as mobile testing units²⁶) regardless of the components most driving the index value in each geographic unit. Also, index weights can be used to inform the types of interventions targeted to specific neighborhoods. In NYC, COVID-19 positive residents could stay in hotels for two weeks until they were no longer contagious²⁷, but it is unclear if this was a targeted intervention. Our index helps contextualize the unique combination of social conditions per ZCTA that relate to ZCTA-level infections for targeting interventions (e.g. in ZCTAs with higher average number of people per household, then rapid testing could be paired with outreach for hotel isolation)."

10) On the first line of page 6 you refer to figure 3 and a comparison figure in the supplement. It would be useful to have both maps as subfigures of figure 3 to ease comparison. It seems that all maps have been generated with different themes and aesthetic choices making direct comparison difficult.

Authors' response: Thank you for this suggestion. We have now removed the supplemental figure maps and included them in Figure 3 so the reader can more easily compare the various maps.

11) It would be worthwhile to perform the same analysis at the ZCTA level to demonstrate that the method and the results are robust. I imagine that individuals at a border may cross the UHF boundary to access other subways as well. In the worst-case scenario, I imagine that there must be some measure of catchment areas of each subway station which may be used to provide greater granularity. This would likely also be more operationally useful.

Authors' response: Thank you for this useful suggestion. We have now modeled this in two additional ways. First, we show a model of subway use by three BWQS risk groups at the UHF level in Supplemental Figure 7. We also show results for three BWQS risk groups at the ZCTA level, as you have suggested, in Supplemental Figure 8. In both cases, we have concluded that they demonstrate the same patterns.

12) Are there any locations where the risk metric does not perform well? If so could you provide more information as to why you think that might be the case and how to best account for it in implementation?

Authors' response: We now provide a residuals analysis (Supplemental Figure 3) which shows that model predictions conform to expectation and there are no large outliers in our model. In fact, it was due, in part, to reviewer comments (including yours) that prompted us to create a more flexible covariate adjustment for the testing ratio per ZCTA (Supplemental Figure 2). Without that adjustment, there were pronounced outliers. We now also provide a map of spatial residuals per Reviewer 1's suggestion (Supplemental Figure 11). We may not appropriately capture population denominators, since there was documented outmigration of more affluent groups, but that would be expected to lead to error primarily in one direction. Finally, we conducted numerous checks for model robustness as prompted by your comment above and found that our BWQS model was very robust.

REVIEWERS' COMMENTS

Reviewer #1 (Remarks to the Author):

The authors have satisfactorily addressed my comments. I have no new comments to add.

Reviewer #5 (Remarks to the Author):

The authors have done an admirable job revising their article. The sections on why and how to use BWQS for social epidemiology is certainly more transparent. I also appreciate the extra leg work it took to compare this method against others and the sensitivity analyses provided. Not necessary to include at all but there is a growing body of literature using other methods which corroborate the impacts of spatial heterogeneity in socioeconomic factors on COVID-19 propagation in New York City. This paper (one that I contributed to) comes to mind immediately (<https://www.nature.com/articles/s41467-020-18271-5>) though it is not nearly as granular as yours. The increased nuance is the discussion around implementation and also adds to the strengths of this paper. I have no other major revisions to suggest.

Response to reviewers

Reviewer 1:

The authors have satisfactorily addressed my comments. I have no new comments to add.

Authors' response: Thank you for your help in improving the quality of our work.

Reviewer 4:

(this reviewer unfortunately did not leave comments in the section for "the authors". However, their comments to the editorial team suggest that they believe the revisions are satisfactory.)

Authors' response: Thank you for your help in improving the quality of our work.

Reviewer 5:

The authors have done an admirable job revising their article. The sections on why and how to use BWQS for social epidemiology is certainly more transparent. I also appreciate the extra leg work it took to compare this method against others and the sensitivity analyses provided. Not necessary to include at all but there is a growing body of literature using other methods which corroborate the impacts of spatial heterogeneity in socioeconomic factors on COVID-19 propagation in New York City. This paper (one that I contributed to) comes to mind immediately (<https://www.nature.com/articles/s41467-020-18271-5>) though it is not nearly as granular as yours. The increased nuance is the discussion around implementation and also adds to the strengths of this paper. I have no other major revisions to suggest.

Authors' response: We have now added a citation to Kissler et al. in our Discussion section as a reference to the other studies associating mobility data with COVID-19 infections. We think it is good context for the reader to understand the work that has been done in this field. Thank you for your help in improving the quality of our work.